

# Avalanche Impact Pressures on Structures with Upstream Pile-Up/Accumulation Zones of Compacted Snow

Perry BARTELT[1], Andrin CAVIEZEL[1], Sandro DEGONDA[2], and Othmar BUSER[1]

[1]WSL Institute for Snow and Avalanche Research SLF, Flüelastrasse 11, 7260 Davos Dorf, Switzerland
[2]ETH Institute for Construction, 8903 Hönggerberg, Zürich, Switzerland

*Correspondence to:* Perry Bartelt (bartelt@slf.ch)

**Abstract.** Existing methods to calculate snow avalanche impact pressures on rigid obstacles are based on the assumption of no upslope pile-up of snow behind the structure at impact. Here we develop a method to predict avalanche impact pressures that accounts for the compaction and accumulation process. We show why this process leads to large impact pressures even at low avalanche approach velocities. The induced pressure depends on the incoming avalanche flow density relative to the ultimate compaction density because this determines the avalanche braking distance and therefore the flow deceleration in the upstream direction. The pile-up/accumulation process induces two additional pressures: (1) the static pressure of the pile-up zone and (2) the tractive stresses operating on the shear planes interfacing the accumulated and still moving avalanche snow. We demonstrate the use of the model on two theoretical examples and one real case study. Avalanche mitigation in maritime regions, or regions undergoing climate change with increasing wet snow avalanche activity, should consider the forces caused by the pile-up/accumulation process in engineering design.

## 1 Introduction

Recent works investigating avalanche-structure interaction have underscored the need to develop better methods to predict avalanche impact pressures (Ousset et al., 2015; De Biagi et al., 2015). There appears to be growing evidence that the long established engineering formula to calculate impact pressure $p(t)$ (time $t$, avalanche flow density $\rho_\Phi(t)$, avalanche velocity $V_\Phi(t)$ shape coefficient $C_D$),

$$p(t) = \frac{1}{2} C_D \rho_\Phi V_\Phi^2(t), \tag{1}$$

is only valid for certain avalanche flow regimes (Sovilla et al., 2008; Baroudi et al., 2011). The formula underpredicts measured values, particularly for slower moving avalanches in plug-flow or "gravitational" regimes (Sovilla et al., 2016). In practice the underprediction is usually compensated by applying non-physical shape coefficients $C_D > 2$.

Equation Eq. 1 is based on two important physical assumptions. The first assumption is that no avalanche mass accumulates behind the structure during the impacting process: When a moving avalanche hits the structure, the smashed snow fragments are assumed to be immediately removed from the impacted surface and re-entrained back into the flow (Bozhinskiy and Losev, 1998). Moreover, avalanching snow is treated as a fluid in which the flux of incoming snow is in balance with the rate of mass





removal at impact. When this condition is satisfied, the application of Eq. 1 is acceptable, i.e. for dry, cohesionless avalanches consisting of disperse agglomerations of snow particles. The formula is correctly applied to model powder avalanche interaction with thin structures, such as trees (Feistl et al., 2014; Bartelt et al., 2018a). It is clearly not valid for slow, dense, cohesive avalanches impacting wide objects where the interaction causes the avalanche to stop or pile-up in front of the structure. That

is, when avalanching snow exhibits some solid behaviour. Many avalanche defense structures – such as dams and other flow obstacles – are purposely designed to induce this process to stop dense flowing avalanches.

The second assumption is of equal importance: The impacted structure is assumed to be perfectly rigid. The structure dissipates none of the incoming flux of kinetic energy in structural deformation energy, but dissipates it entirely at impact in the snow avalanche. This assumption quite often leads to an underestimation of the structural deformations caused by $p(t)$ and

therefore an underprediction of the true internal stress state (Thibert et al., 2008; Baroudi and Thibert, 2009). The application of Eq. 1 must therefore be combined with dynamic magnification factors to account for the impulsive response of the structure when assessing the possibility of structural failure (Clough and Penzien, 1975).

The purpose of this paper is to develop a mechanical model to predict avalanche impact pressure for the case when snow accumulates and piles-up at impact, forming a region of compacted avalanche snow behind the obstacle. The model therefore

accounts for the solid-like behaviour of avalanching snow (Eglit et al., 2007; Faug et al., 2010). We calculate the dynamic impact pressure as a function of the avalanche flow density $\rho_\Phi$ relative to the ultimate compacted solid density $\rho_\Omega$. Avalanche deceleration is calculated based on how kinetic energy is dissipated in the compaction zone. The results reveal why impact pressures in dense plug-flow regimes can be much higher than impact pressures in disperse flow regimes for equal approach velocity. Because we predict the speed of the compaction front, and therefore the loading duration as a function of the incoming

avalanche velocity, the method facilitates the use of dynamic magnification factors in structural analysis.

## 2   Pile-up/Accumulation Impact Pressure

We consider a dense avalanche $\Phi$ impacting a rigid structure $\Upsilon$ with velocity $V_\Phi$, height $h_\Phi$ and bulk density $\rho_\Phi$ (Fig. 1). The structure of width $w_\Upsilon$ is positioned at the position $x=0$; the positive $x$ direction defining the upstream direction of the pile-up. For simplicity we assume the avalanche strikes the structure with a mean depth-averaged velocity and density; that is, both

variables are constant over the flow height but can vary in the streamwise direction and therefore time. For now we assume that the height of the structure $h_\Upsilon$ is higher than the flow $h_\Phi$ i.e. there is no overtopping of the structure. The flux of mass $\dot{M}_\Phi(t)$ and kinetic energy $\dot{K}_\Phi(t)$ arriving at the obstacle and the pile-up zone are therefore

$$\dot{M}_\Phi(t) = \rho_\Phi(t)h_\Phi(t)V_\Phi(t)w_\Upsilon \tag{2}$$

and

$$\dot{K}_\Phi(t) = \frac{1}{2}\dot{M}_\Phi(t)V_\Phi^2(t) = \frac{1}{2}\rho_\Phi(t)h_\Phi(t)V_\Phi^3(t)w_\Upsilon. \tag{3}$$



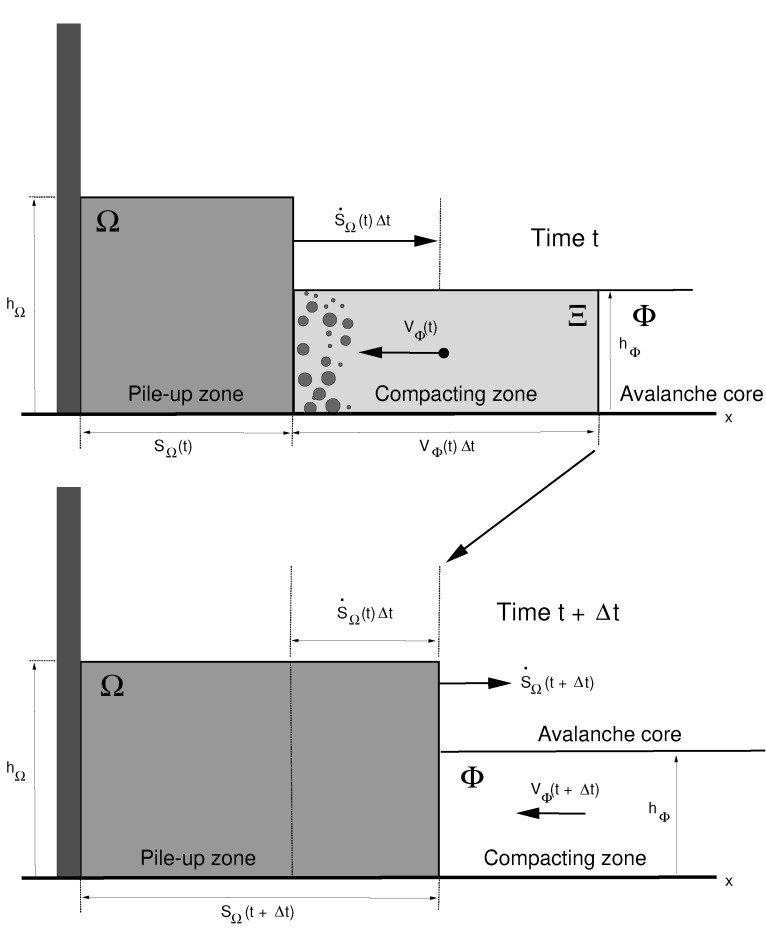

**Figure 1.** Side and plan views of avalanche impact with pile-up and accumulation. The upstream zone is divided into three regions: the dense flowing avalanche $\Phi$, the compacting region $\Xi$ and the pile-up or accumulation zone $\Omega$. The avalanche arrives at time $t$ travelling with the velocity $V_\Phi(t)$, bulk density $\rho_\Phi(t)$ and flow height $h_\Phi(t)$. Within the time interval $\Delta t$ compaction zone $\Xi$ develops in front of the structure with length $V_\Phi(t)\Delta t$. A pile-up zone $\Omega$ with length $S_\Omega$ develops. The pile-up zone is increasing at the speed $\dot{S}_\Omega(t)$. The braking distance of the mass in $\Xi$ is $\Delta d_{\Xi\to\Omega}(t) = \frac{1}{2}\left[V_\Phi(t)\Delta t - \dot{S}_\Omega(t)\Delta t\right]$.




We describe the pile-up process by considering avalanche mass immediately before and after the pile-up. The avalanche is divided into "compacting" avalanche snow (region $\Xi$, density not yet $\rho_\Omega$, velocity not yet zero, time $t$) and "compacted" avalanche snow (region $\Omega$, density $\rho_\Omega$, no velocity, time $t + \Delta t$).

The length of the stationary, compacted, pile-up zone is denoted $S_\Omega$, the height $h_\Omega$. The pile-up height is generally larger

than or equal to the incoming avalanche height $h_\Omega \geq h_\Phi$, but remains smaller than the height of the obstacle $h_\Omega < h_\Upsilon$. The difference between $h_\Phi$ and $h_\Omega$ is a measure of the cohesion. Cohesive flows with strong bonding between the snow clumps have the property $h_\Phi \approx h_\Omega$ (Bartelt et al., 2017a, b). Because of the incoming avalanche, the length of the pile-up zone is growing at the rate $\dot{S}_\Omega$; it is given by conservation of mass,

$$\dot{S}_\Omega(t) = \frac{\rho_\Phi(t)h_\Phi(t)}{\rho_\Omega h_\Omega} V_\Phi(t). \tag{4}$$

During the pile-up, the region $\Xi$ of length $V_\Phi(t)\Delta t$ in the $x$-direction fills in the void space of the compacted zone $\Omega$, see Fig. 1. The difference in the locations defines the braking distance $d_{\Xi \to \Omega}$ over which the incoming mass must stop,

$$\Delta d_{\Xi \to \Omega}(t) = \frac{1}{2}\left[V_\Phi(t)\Delta t - S_\Omega(t)\Delta t\right]. \tag{5}$$

The mean braking speed is therefore

$$\dot{d}_{\Xi \to \Omega}(t) = \frac{1}{2}\left[V_\Phi(t) - \dot{S}_\Omega(t)\right]. \tag{6}$$

The pressure on the obstacle $p_\Xi(t)$ induced by the braking due to the compaction and accumulation is found by equating the work-done by the braking and the flux of incoming kinetic avalanche energy,

$$p_\Xi(t)\dot{d}_{\Xi \to \Omega}(t)h_\Omega w_\Upsilon = \dot{K}_\Phi(t). \tag{7}$$

Therefore,

$$p_\Xi(t) = \frac{h_\Phi(t)V_\Phi(t)}{h_\Omega\left[V_\Phi(t) - \dot{S}_\Omega(t)\right]}\rho_\Phi(t)V_\Phi^2(t) \tag{8}$$

and with the subsitution of the equation for mass conservation

$$p_\Xi(t) = \frac{h_\Phi(t)}{h_\Omega}\left[1 - \frac{\rho_\Phi(t)h_\Phi(t)}{\rho_\Omega h_\Omega}\right]^{-1}\rho_\Phi V_\Phi^2(t). \tag{9}$$

From which it is possible to define an equivalent $C_D$ factor for the pile-up/accumulation regime,

$$C_D = 2\frac{h_\Phi(t)}{h_\Omega}\left[1 - \frac{\rho_\Phi(t)h_\Phi(t)}{\rho_\Omega h_\Omega}\right]^{-1}. \tag{10}$$

Note that the dynamic pressure becomes infinite when $\rho_\Phi h_\Phi = \rho_\Omega h_\Omega$. These values of equivalent $C_D$ are in agreement with

measured values for all $\rho_\Omega > \rho_\Phi$, see Fig. 2, and compare to Sovilla et al. (2008, 2016). This result suggests that impact pressures of slow moving avalanches can be large if the density of the incoming avalanche is near the compaction density.



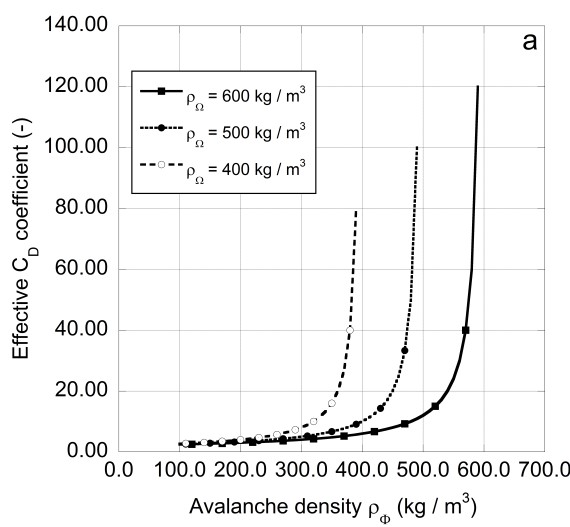
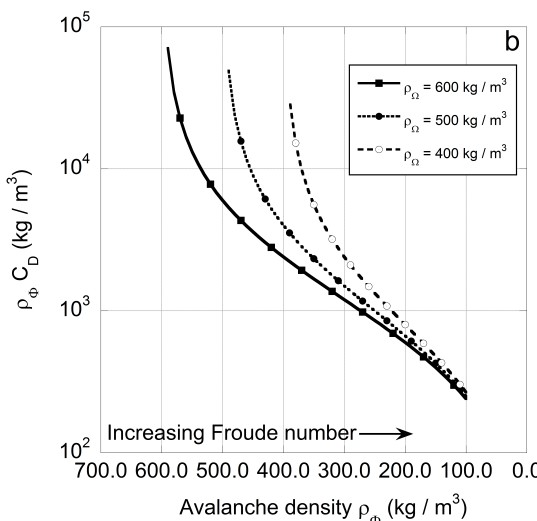

**Figure 2.** a) Effective $C_D$ coefficient (Eq. 10) for different incoming avalanche densities $\rho_\Phi$ and three compaction densities $\rho_\Omega$. The flow height and pile-up heights are equal $h_\Phi = h_\Omega$. Large effective $C_D$ coefficients result when $\rho_\Phi \approx \rho_\Omega$. In this case compacting (braking) distances are short and impact pressures are large. b) The calculated $\rho_\Phi C_D$ are in agreement with values derived from full scale measurements, e.g. (Sovilla et al., 2008). For the sake of comparison to measured values we plot the calculated $\rho_\Phi C_D$ values with decreasing density to mimic increasing Froude numbers (higher Froude numbers correspond to lower flow densities). This produces the effect that effective $C_D$ coefficients are higher for lower flow velocities.

## 3  Mass Accumulation Induces Tractive and Static Pressures on the Obstacle

The total pressure acting on the structure consists of an additional two parts: (1) the static pressure $p_\Omega(t)$ and (2) tractive pressures that develop on the shear planes between the moving and piled-up snow $p_T(t)$, see Fig. 3 and (Faug et al., 2010)

$$p(t) = p_\Xi(t) + p_\Omega(t) + p_T(t). \tag{11}$$

5   All three pressures vary as a function of the accumulation zone $S_\Omega$ and the speed it is growing $\dot{S}_\Omega$. The static pressure of the pile-up zone $\Omega$ is given by

$$p_\Omega(t) = \rho_\Omega S_\Omega(t) \left[ g_x - \mu_\Omega g_z \right] \qquad \text{for} \qquad g_x > \mu_\Omega g_z \tag{12}$$

where $g_x$ and $g_z$ represent the gravitational accelerations in the $x$ and slope perpendicular directions $z$, respectively. The Coulomb parameter $\mu_\Omega$ characterizes the basal friction upstream of the structure. The impact pressure in the pile-up/accumulation

10  regime, unlike the dynamic pressure computed with the standard formula, will depend on the slope angle, as well as the terrain features surrounding the structure. This friction component disappears on a flat slope, $g_x$=0.



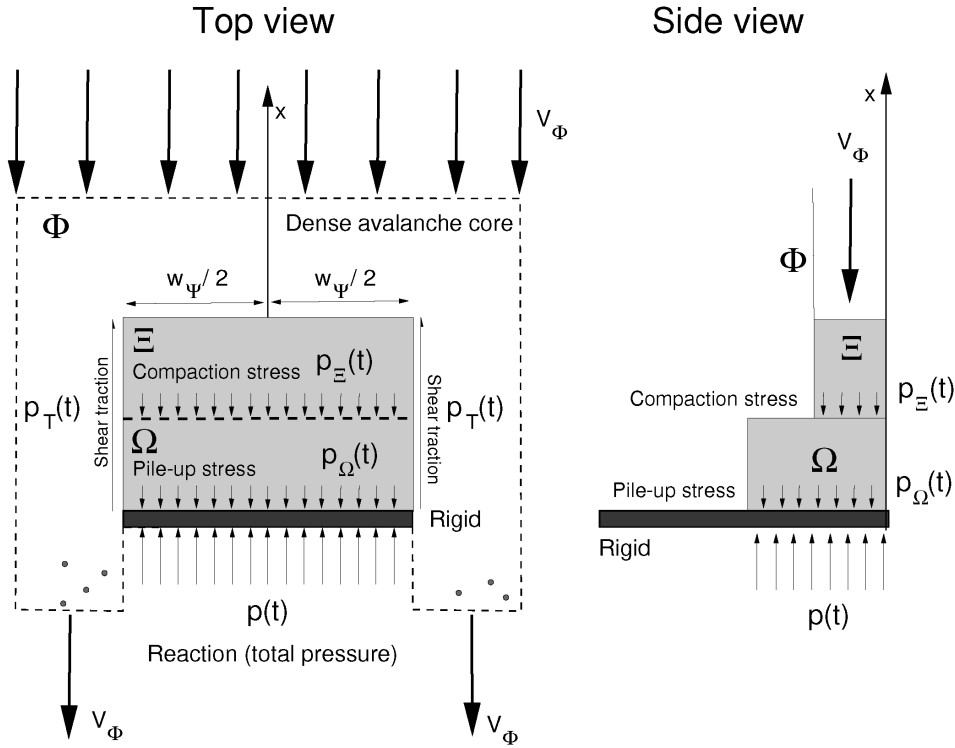

**Figure 3.** Top and side views of avalanche impact with pile-up/accumulation. The upstream zone is divided into three regions: the dense flowing avalanche $\Phi$, the compaction zone $\Xi$ and the pile-up or accumulation zone $\Omega$. The total pressure (reaction) acting on the rigid structure is the sum of the pressures $p(t) = p_\Omega(t) + p_T(t) + p_\Xi(t)$.

On the boundary between the moving and non-moving snow tractive stresses develop. These can only be described by assuming some constitutive relationship between the moving planes, as well as some deformation geometry of the dead zone. For the ideal case of a rectangular dead zone (constant width $w_\Upsilon$), the shear tractions are perpendicular to the structure, requiring no rotation of the shear components into the coordinate system of the obstacle. We emphasize this is not the general case. Assuming we have some velocity-squared drag (parameter $\tau_T$), we can calculate the tractive stresses on the two shear planes, we find,

$$p_T(t) = 2\tau_T \rho_\Phi \frac{S_\Omega(t)}{w_\Upsilon} V_\Phi^2(t). \tag{13}$$

The dimensionless shearing plane resistance $\tau_T$ can be well approximated by the Voellmy formula $\tau_T \approx g/\xi$, where here $\xi$ is the Voellmy velocity squared drag coeffficient.





## 4   Applications

### 4.1   Constant Velocity: $V_\Phi(t)$ = 10 m/s

As a first example we calculated a 15 s long impact with a constant incoming avalanche velocity $V_\Phi(t)$ = 10 m/s. This velocity is often used to separate the red and blue danger zones in hazard mapping applications in Switzerland. The calculated pressure with standard equation Eq. 1 predicts a pressure value of $p(t)$ = 30 kPa for flow density $\rho_\Phi$ = 300 kg/m$^3$ and $C_D$ = 2. We do not consider static $p_\Omega(t)$=0 or tractive $p_T(t)$=0 contributions to the pile-up pressure.

In the first series of calculations we set $h_\Phi = h_\Omega$ and determined the impact pressure as a function of the avalanche flow density $\rho_\Phi$ but with different compaction densities $\rho_\Omega$ (400 kg/m$^3$, 500 kg/m$^3$ and 600 kg/m$^3$), see Fig. 4a. Large impact pressures are associated with higher densities relative to the compaction density. The larger the compaction density, the slower the rise in calculated pressure. Moreover, the compaction density plays an important role in determining the magnitude of the impact pressure.

In a second series of calculations we allowed some increase in the pile-up height. That is, we modelled a less cohesive flow. For comparison we set $h_\Phi$ = 2.0 m and $h_\Omega$ = 2.5 m, see Fig. 4b. Here too, presssures greater than 30 kPa can be found for higher flow densities. Pressures greater than 100 kPa require both high flow density $\rho_\Phi$, but also low compaction densities $\rho_\Omega$. Impact pressures over 100 kPa can be expected for $V_\Phi(t)$ = 10 m/s for flow densities $\rho_\Phi \approx$ 400 kg/m$^3$.

### 4.2   Variable Velocity $V_\Phi(t)$ and tractive stresses $p_T(t)$

To demonstrate how the model predicts impact pressure when the avalanche flow velocity and density vary over time we consider two examples.

In the first example an avalanche impacts a structure with an approach velocity of $V_\Phi(0)$ = 26 m/s; the velocity decreases in time to approximatly $V_\Phi(t > 9s)$ = 10 m/s (Fig. 5a). The high velocity region lasts only several seconds. We assume a relatively low front density $\rho_\Phi$=220 kg/m$^3$. The flow density increases towards the avalanche tail. The flow height of the incoming avalanche is constant $h_\Phi$ = 3m; the pile-up height is equal $h_\Omega$ = 3m. We assume a pile-up density of $\rho_\Omega$=500 kg/m$^3$. Figure 5b compares the calculated pressures $p_\Xi(t)$ and $p_\Xi(t) + p_T(t)$ with the pressure of the standard engineering formula Eq. 1 (red line, $C_D$ = 2). We take $\tau_T \approx$ g/$\xi$, with $\xi$ = 2000 m/s$^2$ and assume no static pressure $p_\Omega(t)$=0. Here the pile-up formula predicts pressures approximately twice as large as the standard formula, for both the front and tail of the avalanche. The calculated equivalent $C_D$ values are between 3 and 5 for this example.

In the second example a high density $\rho_\Phi(t) > 450$ kg/m$^3$, slow moving avalanche $V_\Phi(t)$ = 2.5 m/s with large flow height $h_\Phi(t)$ = 5.00 m strikes a rigid obstacle, Fig. 6a. We consider three cases $h_\Omega$ = 5.50 m, $h_\Omega$ = 5.25 m and $h_\Omega$ = 5.00 m. The calculated impact pressures are shown in Fig. 6b and compared to the standard impact formula (red line, $C_D$ = 2). In this example we see the clear role of cohesion in the avalanche core (Bartelt et al., 2017a). For highly cohesive flows there is no pile-up extension in the height $h_\Phi(t) = h_\Omega(t)$ and the impact pressures exceed 100 kPa for a low impact velocity. The equivalent $C_D$ coefficients for this case is approximately 50. The pressures are significantly smaller when the flows are





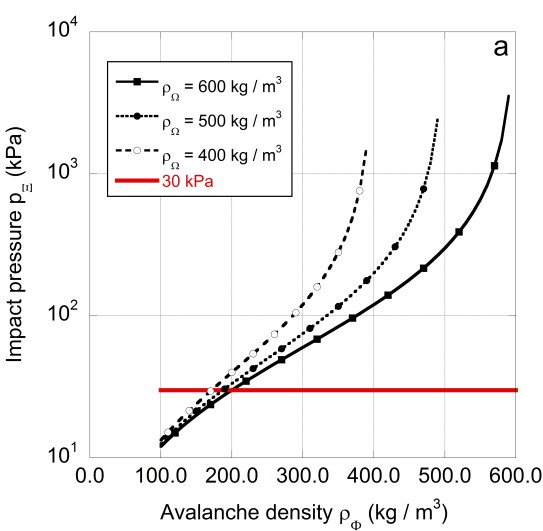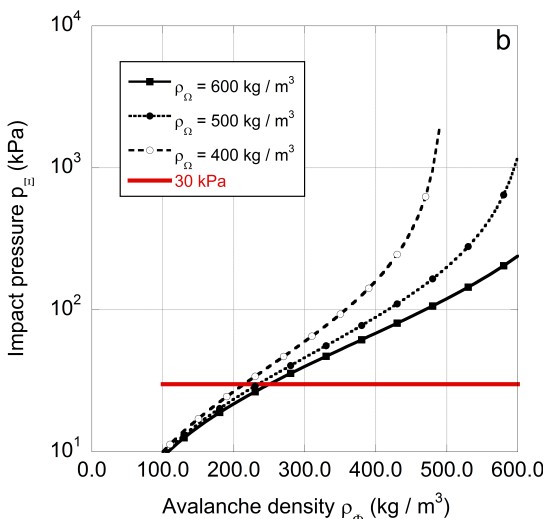

**Figure 4.** Caclulated impact pressure for constant avalanche flow velocity $V_\Phi = 10$ m/s, different flow densities $\rho_\Phi$ and three compacted densities $\rho_\Omega$. The standard equation predicts an impact pressure of $p(t) = 30$ kPa (red line) a) $h_\Phi = h_\Omega$. b) $h_\Phi = 2.0$ m and $h_\Omega = 2.5$ m. For flow densities near the compaction density, the pressures are higher than givne by Eq. 1.

cohesionless $h_\Omega > h_\Phi$. In this case the equivalent $C_D$ coefficients drop to 10. Clearly, the pressures decrease rapidly when $h_\Omega(t) > h_\Phi(t)$. This result underscores the important role of snow quality in the flowing avalanche at the time of impact.

Although we considered a tractive stress in this example, the tractive stresses where small in comparison to the dynamic pressures $p_T(t) < p_\Xi(t)$. The flowing avalanche snow does not have a high enough velocity to exert large tractive stresses on the sidewalls of the pile-up zone.

Finally, these two case studies were motivated by impact pressures reported in Sovilla et al. (2016). Thus, a comparison to measured values is possible (we obtain values close to the measurements), However, to apply the model we must make assumptions regarding the flow and compaction densities, as well as the snow quality (possible pile-up height). This data is simply not available (and might never be available). Because measurement data seldom contains information of the upstream pile-up process, including the propagation speed of the compaction front, a direct comparison to measured pressures is at present helpful, but certainly not conclusive. The examples should motivate experimentalists to capture more upstream data of the stopping process, especially compaction densities and the extent of the dead-zone.

### 4.3 Damaged bridge, Mittelbedra Avalanche, Davos

In the final example we demonstrate a salient application feature of the pile-up/accumulation model: To apply it we need to predict the streamwise variation in avalanche flow density. Moreover, the model cannot be applied in conjunction with constant



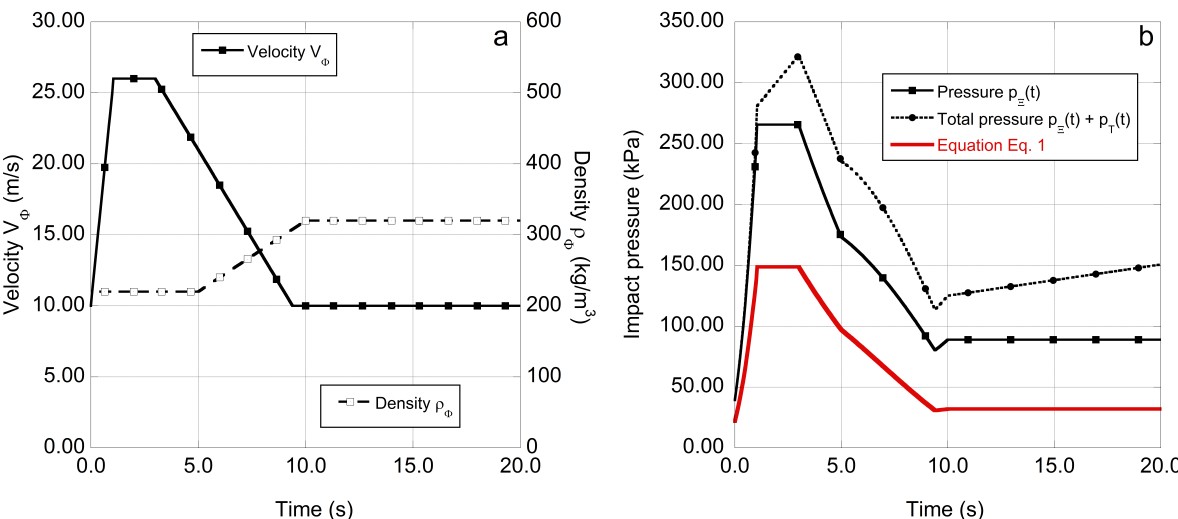

**Figure 5.** a) Incoming avalanche velocity $V_\Phi$ and flow density $\rho_\Phi$. b) Calculated impact pressure in the pile-up/accumulation regime considering only the impulsive pressure $p_\Xi$ for an avalanche with incoming velocity of $V_\Phi(0) = 25$ m/s. Comparision to standard calculation formula, no traction $p_T = 0$.

density avalanche dynamics models because these models do not provide any measure of the incoming flow density. Without the flow density it is impossible to find the upstream braking distance and therefore the force acting on the obstacle.

An interesting case study presented itself in January 2018 when a flowing avalanche with model flow volume (30,000 m$^3$) struck a highway bridge located near Mittelbedra on Flüelapass road near Davos, Switzerland (Fig. 7). The avalanche destroyed

5    the bridge guardrail over a length of 20m. This is somewhat unusual, since bridge guardrails in Switzerland are designed to withstand betweem 100 kN (10 ton) vehicle impact loadings. These correspond to impact pressures of approximately 100 kPa. The engineering question subsequently arose how could the Mittelbedra avalanche induce such large impact loadings.

Field examinations with drone flights identified the position and size of the avalanche release zone (Fig. 7). Two small slabs released (average release height 1.0m; release volume 10,000 m$^3$; release density 300 kg/m$^3$ elevation 2100 m). The avalanche

10   descended down a steep gully, entraining snow, before hitting the bridge (elevation 1720 m). The avalanche came to rest in the stream at the valley bottom, but much snow was piled-up on the bridge, indicating the possibilty of some snow accumulation. The piled-up snow was immediately removed by road operation crews.

The field information was used to define the initial and boundary conditions for avalanche dynamics calculations (Christen et al., 2010). A model was applied that predicts streamwise variations in avalanche flow densities (Buser and Bartelt, 2009, 2015;

15   Bartelt et al., 2016). The model was able to reproduce the observed flow path, flow width at the bridge as well as depositions in the stream (Fig. 7). The calculations (using a 2.5 m x 2.5 m grid resolution) indicated that the avalanche was travelling at



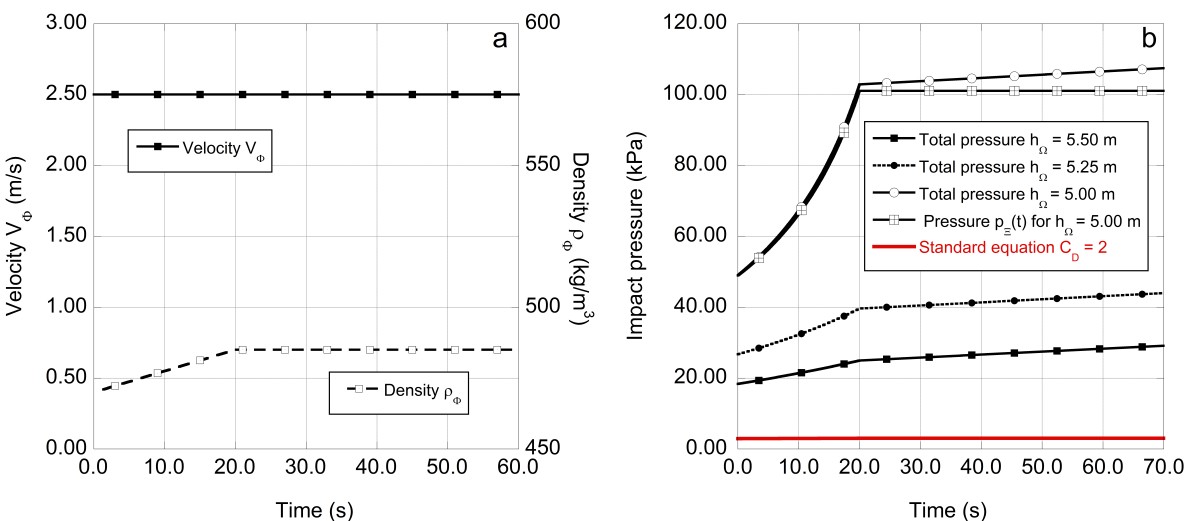

**Figure 6.** a) Incoming avalanche velocity $V_\Phi$ and flow density $\rho_\Phi$ for a dense slow moving avalanche. Avalanche flow height $h_\Phi = 5.00$ m. b) Calculated impact pressure in the pile-up/accumulation regime for different pile up heights $h_\Omega = 5.50$ m, $h_\Omega = 5.25$ m and $h_\Omega = 5.00$ m. Comparison to standard calculation formula (red line).

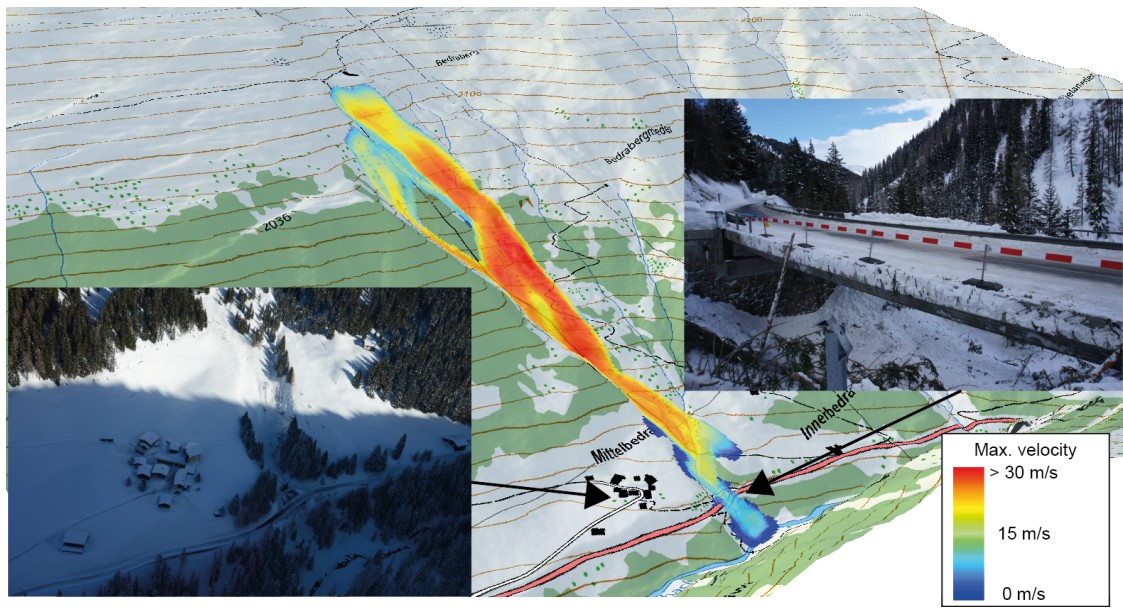

**Figure 7.** Mittelbedra avalanche, Davos. A flowing avalanche struck the highway bridge removing the side guardrail. Avalanche dynamics calculations were performed to estimate the avalanche arrival velocity (12 m/s).





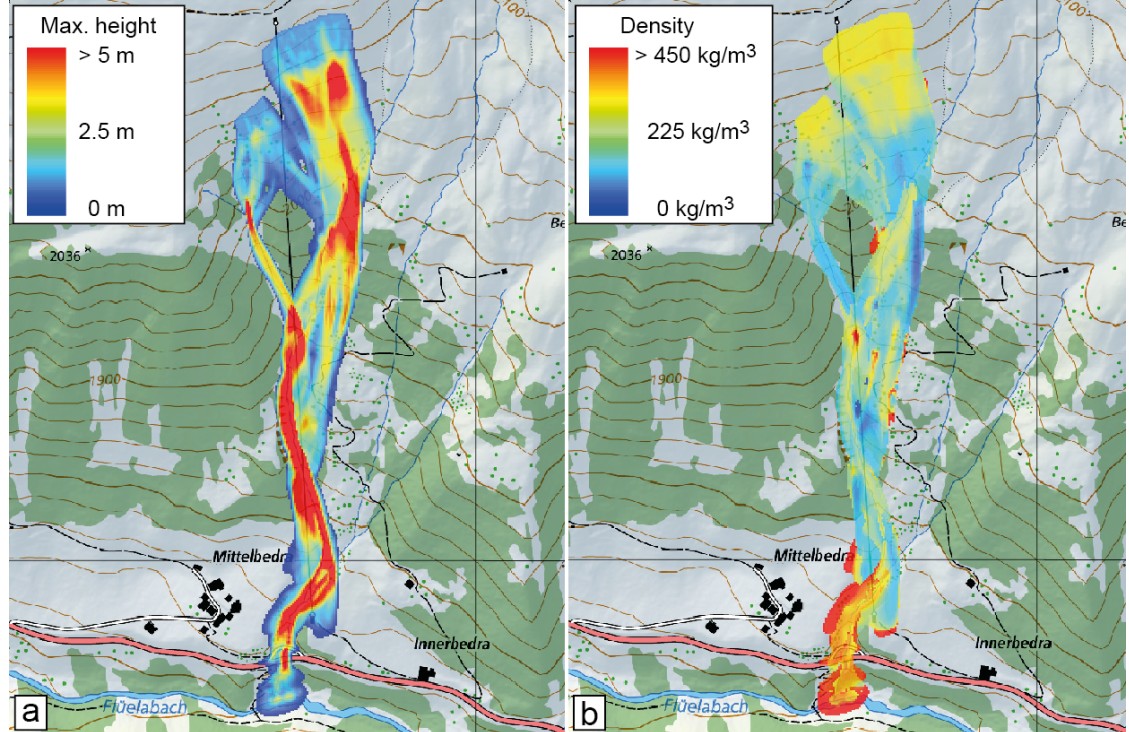

**Figure 8.** Calculated a) flow height and b) flow density of the Mittelbedra avalanche (Davos, Switzerland). The calculated flow heights at the bridge are large enough to fill the overboard such that mass piles-up on the bridge. Although the avalanche was fluidized on much of the avalanche path, calculated avalanche flow densities at the bridge are approximately $\rho_\Phi$

approximately 12 m/s when it struck the bridge, with a (fluidized) flow height of over 5m, see Fig. 8a (total avalanche volume at impact 30,000 m$^3$). This would be enough to fill the bridge overboard. The simulations also revealed that the avalanche was fluidized for much of its flow duration (Fig. 8b), but densities at impact were approximately $\rho_\Phi = 300$ kg/m$^3$. These flow densities are considered to be smaller than a compaction zone density. A possible loading case would therefore be that much of

5  the avalanche mass flowed under the bridge, but the upper regions of the flow hit the bridge, began to pile-up, but then because of the high pressures, the guardrail failed and was sweept away by the flow.

We caclculated the impact pressure by specifying the simulated velocity time history $V_\Phi(t)$ and streamwise density history $\rho_\Phi(t)$, see Fig. 9a. Assuming that some avalanche mass flowed under the bridge we specified $h_\Phi = 2$ m. We varied the pile-up height at the guardrail: $h_\Omega = 2.0$ m, $h_\Omega = 2.5$ m and $h_\Omega = 3.0$ m. We therefore considered a highly cohesive and two

10  non-cohesive flow regimes. The calculated impact pressures are shown in Fig. 9b and compared to the standard formula (red line), $C_D = 2$. We find peak impact pressures over 250 kPa (25 tons/point loading) for the cohesive impact(Fig. 9b), clearly in excess of the standard pressure formula (50 kPa). This impact pressure is deemed too high; but less cohesive flow regimes can provide more reasonable impact pressures on the order of 100 kPa (10 tons/point loading). Calculated impact pressures using the standard formula are lower (max 50 kPa).





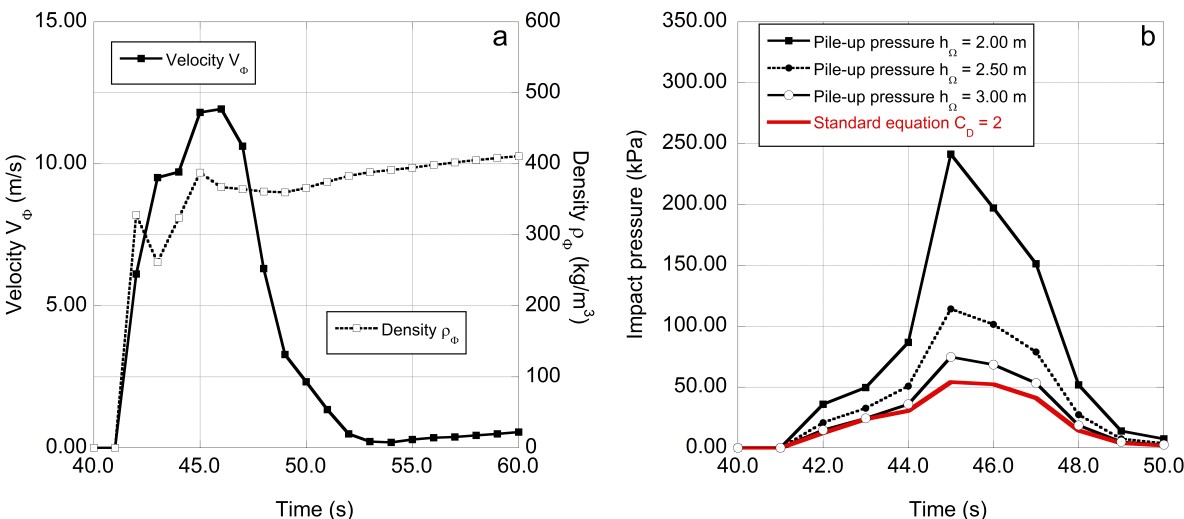

**Figure 9.** Mittelbeda avalanche, Davos. a) Calculated avalanche arrival velocity $V_\Phi(t)$ and density $\rho_\Phi(t)$ b) For a cohesive flow $h_\Phi = h_\Omega$, the calculated peak pressure is more than five times the value predicted by the standard formula (red line). Similar results to the standard formula are obtained when the pile-up height is $h_\Omega = 3.0$ m. Compaction density: $\rho_\Omega$=500 kg/m$^3$.

We emphasize that the investigated pile-up/accumulation loading regime is one possible scenario and demostrates the dangers of low avalanche approach velocities where solid pile-up is possible. The method should not be applied randomly to all case studies, but only those where mass accumulation is deemed possible.

## 5   Conclusions

5   The determination of impact pressures on rigid obstacles remains a difficult engineering problem. Here we have provided a mechanical description, based on the assumption of the compaction and densification of avalanche snow, why impact pressures can indeed be large, much larger than expected by standard engineering formulas. The condition to create such large dynamic pressures for slow moving flows is the formation of a pile-up/accumulation zone at the upstream face of the impacted obstacle. In this case the induced pressure $p_\Xi$ cannot be represented by the Froude number, rather the ratio of the density of the incoming
10   snow $\rho_\Phi$ to the ultimate compacted snow density $\rho_\Omega$. We find for $h_\Phi = h_\Omega$,

$$p_\Xi(t) = \left[1 - \frac{\rho_\Phi}{\rho_\Omega}\right]^{-1} \rho_\Phi V_\Phi^2(t). \tag{14}$$

High density flows (that is, slow moving flows) will exert large pressures when the snow cannot be compacted. In fact, in the theoretical case $\rho_\Phi = \rho_\Omega$, the impact pressure is infinite, because the braking distance reduces to zero. The braking distance,





and therefore the magnitude of the force exerted on the obstacle, is directly related to the densification of the avalanching snow. Fortunately avalanche snow is a compactible material and the impact pressures are finite. Increasing the pile-up height $h_\Omega$ will also reduce the applied pressure. Thus, cohesive flows which exhibit strong material bonding will exert higher impact pressures on structures. The explanation why flows with low Froude numbers exhibit correspondingly higher pressures is that these flows

are simply denser, and their slow movement facilitates the formation of a pile-up zone. It is reassuring that the equivalent $C_D$ values we calculate

$$C_D = 2\frac{h_\Phi}{h_\Omega}\left[1 - \frac{\rho_\Phi h_\Phi(t)}{\rho_\Omega h_\Omega}\right]^{-1}. \tag{15}$$

are directly comparable to values derived from experimental observations (Sovilla et al., 2008, 2016). Avalanche dynamics models will need to predict streamwise variations in avalanche flow density in order to calculate impact pressures for the

pile-up/accumulation regime (Buser and Bartelt, 2015; Bartelt et al., 2016).

Another important conclusion we make from our analysis is that when avalanche mass accumulates behind structures the impact pressure $p(t)$ can be generally expressed as a linear combination of three components,

$$p(t) = p_\Omega(t) + p_T(t) + p_\Xi(t). \tag{16}$$

These components are all interrelated by the geometry of the dead zone which defines both the magnitude of the static pressure

$p_\Omega(t)$ as well as the location of the shearing interfaces and therefore the reaction to the tractive pressures $p_T(t)$. In this paper we have considered only one possible geometry: a backfill zone of constant width $w_\Upsilon$ equal to the width of the impacted structure. Our analysis therefore reveals that the total pressure $p(t)$ in the backfill regime is a complex function of many time-dependent parameters (e.g. incoming avalanche velocity and density) as well as time-independent material parameters describing avalanche snow, specifically the compaction density $\rho_\Omega$, the friction in the pile-up zone $\mu_\Omega$ and the tractive friction

on the shear planes $\tau_T$. We purposely limited the physical description of each pressure component $(p_\Omega(t), p_T(t), p_\Xi(t))$ to a *single* constitutive parameter for each process $(\mu_\Omega, \xi_T, \rho_\Omega)$. Moreover, even the most simple pressure calculations will require engineers to assume some displacement configuration of the backfill process. This will be difficult, see (Faug et al., 2010) for the example of wedge shaped back-fill regions.

Our final conclusion underscores the limits of on-going field investigations. Reports of experimental measurements typically

present pressures $p(t)$ but no information concerning the pile-up process. It is therefore impossible to use existing field measurements to validate pile-up/accumulation pressure models and their associated parameters. Empirical formulas of the form

$$p(t) = A(h_\Omega, \mu_\Omega) + \frac{1}{2}C_D\rho_\Phi V_\Phi(t)^2 \tag{17}$$

do not physically represent the complex stopping process dominated by the spreading of compaction fronts and intense strain

localization induced during the formation of shearing interfaces between the moving and stopped avalanche snow. Such formulas, which implicitly assume the dynamic stresses are characterized by no accumulation, yet likewise invoke assumptions of hydrostatic stress-states, should be applied with extreme caution, or outright rejected because they are divorced from a specific deformation mechanism and therefore a specific loading process. Finally, we assume the backfill pressure arises from the




interaction with a rigid structure. To determine the internal stresses within the structure (i.e. failure), dynamic magnification factors must be found which depend on the stiffness and mass distribution of the impacted body (Bartelt et al., 2018a). The fact that pressure sensors are mounted on flexible structures should be considered in the analysis of the experimental results, e.g. (Thibert et al., 2008; Baroudi and Thibert, 2009). We hope our contribution will serve as a guide to define what information

5   should be gathered in future field experiments. Of considerable importance are simple density measurements of compacted snow $\rho_\Omega$ in pile-up zones.

*Acknowledgements.*  This work was performed within the framework of the joint Austrian-Swiss project bDFA, a study of avalanche motion beyond the dense flow avalanche regime. We thank the Austrian Academy of Science (ÖAW) for their financial support as well as the Austrian research partners (Austrain Research Centre for Forests, Torrent and Avalanche Control and the Universtiy of Innsbruck). Further support is

10   provided by the on-going WSL research program Climate Change and Mass Movements (CCAMM).



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
