# Peer review of "Avalanche Impact Pressures on Structures with Upstream Pile-Up/Accumulation Zones of Compacted Snow"

_Natural Hazards and Earth System Sciences, 2018_

## Short Comment (SC1) · 24 Sep 2018

The authors pick up an interesting topic about the impact pressure of avalanches on rigid obstacles. However, although the first author is a re-known avalanche researcher, he seems to be not aware that the topic was already discussed and summarized in:

Gauer, P. & Jóhannesson, T. (2009) in Jóhannesson, T.; Gauer, P.; Issler, D. & Lied, K. (Eds.)  The design of avalanche protection dams.  Recent practical and theoretical developments Chapter 11 Loads on walls European Commission, Directorate-General for Research, 2009 , 77-94

[Figure]

Hence, the authors' approach does not really present originality.

---

## Author Comment (AC1) · 26 Sep 2018

Dear Peter,

Many thanks for your comment. We formulated our approach using papers discussing field scale impact experiments. We find it remarkable that nobody has applied your results in the last 10 years to evaluate measured impact pressures.

We developed our model using a completely different approach: Application of the work-energy theorem. We get comparable results. However, our approach leads to further insights into avalanche impact, that is not discussed in Chapter 11. These

differences in derivation become important in understanding the overall problem.

The work theorem states that the change in kinetic energy $\Delta K$ is equal to the product of the force F (on the wall from pile-up) and the braking distance d of the avalanche mass stopped by the wall,

$\Delta K = F\ d$.

Using the standard definition for impact pressure $p = F/A$

$p = 1/2\ C\_d\ \ U\hat{}2$

(where $C\_d$ = stress intensity factor; = density; U = approach velocity, A = cross-sectional area) we arrive at an interesting re-definition of the stress intensity factor:

$C\_d = (V/A)/d = l/d$

where V is the volume stopped by the wall, with ($l = V/A$). Interestingly, the stress intensity factor (for pile-up) is simply defined as the length of incoming mass scaled to the braking distance. Of course, this is a crude approximation (we don't consider potential energy, true for an avalanche moving on a plane); however, it is equivalent to your treatment. We both take $l = U\Delta t$. For us, the braking distance is given by the compacted density $\_0$ of the snow and mass conservation

$d = 1/2\ (U - S\ \grave{I}\breve{G}\ )\Delta t = 1/2\ (1 - /\_0\ )U\Delta t$.

The advantage of the work-energy approach is that we can divide up the incoming mass into a pile-up mass $l(t)$ and a deflected mass (or splashed mass). Moreover, mass that is not piled-up, must be deflected. That is, $l \neq U\Delta t$. In fact, we can define a $l(t)$ that depends on the geometry of the pile-up zone – for example, wedges. For the engineer, defining $l(t)$ then becomes the key problem in determining appropriate avalanche impact pressures. Knowing the pile-up geometry, we can predict the duration time of the pile-up force – the peaks in pressure and therefore dynamic magnification factors. An important aspect is that the time to pile-up is very short compared

to the time the avalanche needs to flow by. Thus, in the general case we are confronted with two sources for the impact force – pile-up and deflection. This is true for both walls and thin objects. (Deflection requires invoking Newton's laws of changing momentum to find the force.)

The approach of invoking the work-energy theorem thus opens many doors to consider different impact situations, including the interpretation of experimental results. Frankly, it is impossible to interpret experimental measurements without knowing I(t). We think that it is very important for practicing avalanche engineers to be presented with basic, compelling and consistent explanations. Although your chapter is of great interest to us for several reasons, we intend to follow a different path and see where it leads.

---

## Referee Comment (RC1) · Anonymous Referee #1 · 9 Oct 2018

The discussion paper titled "Avalanche Impact Pressures on Structures with Upstream Pile-Up/Accumulation Zones of Compacted Snow" by Bartelt *et al.* proposes a very interesting approach for dealing with the interaction between snow avalanche flow and isolated obstacles. The model considers that the deceleration of the snow mass turns into impact pressure against the obstacle. The paper clearly states its limitations: in particular, the considered geometry is simple, and an ideal rectangular dead zone is assumed (which is not real for narrow obstacles). In addition, the obstacle is supposed rigid, thus the dynamic effects are not considered. Nevertheless, the proposed mechanical model is worthy of attention, with particular reference to the engineering problems in mountain areas.

[Figure]

**Specific comments**

1. The authors state that cohesive flows with strong bonding between the snow clumps have the property $h_\Phi \approx h_\Omega$. In this case, a compaction of the impacting mass occurs, rather than a pile-up. A short comment is probably expected.

2. Referring to the analytical model, as well detailed in the discussion paper, the region $\Xi$ has length $V_\Phi(t)\Delta t$, while the resulting pile-up zone has length $\dot{S}_\Omega(t)\Delta t$. The authors indicate the braking distance as $d_{\Xi \to \Omega}$ (p.4 line 11). From the sketch in Figure 1, it results that the mean braking distance is the distance between the centers of mass of the compacting and the pile-up zones, i.e.

$$d_{\Xi \to \Omega}(t) = \frac{1}{2}\left[V_\Phi(t)\Delta t - \dot{S}_\Omega(t)\Delta t\right].$$

Why do the authors adopt a different symbol for the braking distance in Eqn. (5), i.e. $\Delta d_{\Xi \to \Omega}$? It is expected that $\Delta d_{\Xi \to \Omega}$ is the variation of the braking distance at different times, say $t$ and $t + \Delta t$. In addition, the authors should also clarify what do they intend with $\dot{d}_{\Xi \to \Omega}(t)$. It is expected that this term is the time derivative of the braking distance, i.e.,

$$\dot{d}_{\Xi \to \Omega}(t) = \lim_{\Delta t \to 0} \frac{\Delta d_{\Xi \to \Omega}}{\Delta t} = \lim_{\Delta t \to 0} \frac{d_{\Xi \to \Omega}(t + \Delta t) - d_{\Xi \to \Omega}(t)}{\Delta t}.$$

Can the author better explain what do they intend with braking speed? Is it the ratio between the braking distance and $\Delta t$? Probably, it would be better to indicate the braking speed with a symbol without the dot.

3. Observing Figure 3, it seems that the shear traction force is directed against the snow avalanche flow, i.e., a negative pressure is acting on the obstacle. Have I well understood?

4. Can the author include some references about the lateral requirements resistance of bridge guardrails? European norms (say EN 1317) relate to performance classes based on impact speed, angle and vehicle mass, rather than impact loadings.

5. Limiting the attention to the failure of the guardrail, any impact pressure larger than the one that caused the observed damage would cause the same damage. However, the presence of further elements that were not destroyed by the avalanche can help in estimating an upper limit of the impact pressure. Have the authors found other elements that can help in estimating an upper limit of the impact pressure?

**Minor observations**

- $\dot{S}_\Omega$ in Eqn. (5)

- The paper "Formation of levees and en-echelon shear planes during snow avalanche run-out" by Bartelt *et al.* dates back to 2012, rather than 2017.

---

## Referee Comment (RC2) · Anonymous Referee #2 · 15 Oct 2018

The manuscript "Avalanche Impact Pressures on Structures with Upstream Pile-Up/Accumulation Zones of Compacted Snow" by Bartelt et al. discusses an approach to calculate/estimate pressures arising from avalanches hitting stationary obstacles. It presents an interesting mechanical model, the theoretical basis thereof, applied examples and states the limits and shortfalls. The setup of the paper is clear and follows a logical structure. The model seems to be ready to be included in dynamical avalanche models once the necessary basic changes (like variable densities) are implemented. This presents a very promising and needed approach to discuss impact pressures on obstacles.

Main comments:

- The compaction density Rho_omega lacks the necessary discussion. Since it is one of the main driving factors for the results (e.g. fig 2 / 4) it needs a better justification. Rho_omega is currently not (yet) available from models, and reliable observations are not easily accessible. The other parameters of this approach can be handled by models or observations. So how do the authors suggest to handle this important (tuning) parameter? In the paper the model is sometimes tested with three (arbitrary, if plausible) densities, and sometimes set to a fixed value (e.g. label fig 9). Especially in the case for the Mittelbeda avalanche it is (seemingly) picked at random. This needs to be better justified with observations, or at least the reasoning for this specific value needs to be shown.

- Figures 1 and 3 need to be improved. Figure 1 has separate compacting zone CZ and avalanche core AC in the upper panel. Then in the lower panel CZ and AC are the same, however the v is denoted with t + delta_t. I somehow expect there to be a CZ(t + delta_t) and the same for AC. Or if the authors try to show the "steady" state reached at end of compaction, remove the CZ in the lower panel and make it more clear in the label. The right panel of figure 3 contains basically the same (simplified) information as figure 1, adding only information about stress. Presenting the side view in the same manner as the top view leads to confusion. I suggest either to include the information about stress in fig 1 and reference it, or rotate the right panel of fig 3 by 90 deg to make the "side view" clearer.

Comments:

- On p.4 / l. 4 it is stated "The pile-up height is generally...". How do the authors come to this conclusion? On p.2 /l. 25 part of this is presented as an assumption...

- I suggest moving sentence p.5 / l. 9-10 to the beginning paragraph of section 2. This would be beneficial to the reader wondering about other influencing factors right from the start of the discussion.

[Figure]

- Regarding SC1 by Peter Gauer: I suggest including a short remark about the discussed work in the introduction.

- For easier readability I suggest moving p. 8 / l. 6-12 to the beginning of the section. The information that both cases "are motivated by... " observations is an important one.

- P 9. / l. 5: Remove "somewhat". Very unspecific: either it is unusual or it is not unusual.

- Out of interest: what causes the drop in velocity in fig. 9. a) at approx. 42 seconds?

Minor:

- Label fig. 4: givne -> given

- P 9. / l. 6: betweem -> between

- P 9. / l. 9: "at" missing between density and elevation

- P 9. / l. 11: possibilty -> possibility

All (obvious) typos are in section 4.3, seems to be avoidable by autocorrect.

---

## Referee Comment (RC3) · T. Faug (Referee) · 19 Oct 2018

**Review of the paper by P. Bartelt et al., entitled "Avalanche Impact Pressures on Structures with Upstream Pile-Up/Accumulation Zones of Compacted Snow"**

October 19, 2018

**1 General comment**

The topic addressed by the authors is of utmost importance. The calculation of the impact force of avalanches on obstacles when a fluid-to-solid transition occurs, thus forming stagnant (quasi-static) zone upstream of obstacles and traveling jumps, is a challenging question. Although a number of significant advances were made in the recent years (most of them are available along my report below), there remains a lot to do because of the complicated physics which takes place during dense flow/obstacle interaction. The present paper proposes an approach based on a (simple) *"energy approach"*.

I read in detail the ideas developed by the authors. I must say that I have a number of major concerns about the theoretical part proposed by Perry Bartelt et al. The main reasons are (at least) the following:

The paper is –first of all– not free of misconceptions:

- the approach proposed starts with some equations that are pulled out of a hat (Eq. (3) for instance) or even wrong (Eq. (4)); this poses a serious problem because all the results presented in the rest of paper (virtual cases and practical case) depend on those confusing or flawed equations stated at the beginning of the description of the model.

- I also found a couple of misleading statements (see the section **specific comments** below).

Moreover, I must say that the present study does ignore a number of important works done before on the topic. I thus fully agree with the short comment earlier proposed by Peter Gauer on this weak point of the paper.

Unlike referees 1 and 2, I can not provide a positive feedback on the present study (see section **Recommendation** at the end of the present report).

I have taken time to write a rather detailed review in order to explain where the outcomes of the previous studies were relevant and could (not to say should) have been considered. I really invite the authors to read and consider the efforts made in the recent years by other researchers on the topic. In particular, they should demonstrate that their energy approach (ONCE CORRECTED) is superior to previous approaches mostly based on momentum conservation equation. At this stage, I did not go through the details of the examples/applications (section 4) because some equations used to draw the different plots and presented at the beginning of the paper are flawed.

**2   Specific comments**

- abstract: *"Existing methods to calculate snow avalanche impact pressures on rigid obstacles are based on the assumption of no upslope pile-up of snow behind the structure at impact."* This sentence shall help the author to promot their model but it must be removed because this is a wrong statement! There are methods—already published—that make effort to address carefully the problem and the authors should not ignore them: see the references cited along my review comments below.

- page 1, lines 18-19: why using this term "shape coefficient" for $C_D$? In fluid mechanics $C_D$ is the "drag coefficient". Here, the EU handbook edited by Tomas Johannesson et al. (2009) would merit citation. In particular, table 12.1 (page 107) provides recommendations for the values of $C_D$.

- page 2, lines 3-5: it is well-established that when granular flows impact walls, the formation of dead zones upstream of the obstacle is a key process that we need to include into the impact force calculation in order to predict the correct impact forces. There are a number of published papers on the problem that would merit to be mentioned here: please see (Faug et al., Phys. Rev E 2009) for walls overtopped by a steady granular flows, see (Chanut et al., Phys. Rev. E 2010; Faug et al., Phys. Rev. E 2011) for walls overtopped by a transient granular flows, and more recently Albaba et al. (Phys. Rev. E 2018) for semi-infinite rigid walls (no overtopping). Moreover, it is also a well-established fact for snow avalanches/obstacle interaction problems that dead zones and shock-waves traveling upstream are important physical processes: see, and please cite, the EU Handbook 2009 (already mentioned above), and Faug et al. (Ann. Glaciol 2010). Also some other papers on the topic by B. Sovilla and co-workers, as well as by Shiva Pudasaini and co-workers would merit much more attention.

- page 2, line 3 again: note that "cohesive avalanches" is not a necessary condition. Dry (cohesionless) granular flows also produce dead zones at the impact with wide obstacles (see the literature mentioned above).

- figure 1: notation used here and all along the paper is weird... your drawing is nothing else than a traveling jump that is classically observed in water and granular flows (and snow avalanches) when those flows transition from a supercritical to a subcritical flow regime (for instance when those flows impact a wall). The difference between water flows and granular (or snow) flows is the fact that the granular (and snow) jumps may be compressible and accompanied by a shock in density, in addition to both velocity and height discontinuities. In general, in fluid mechanics the notation used is $h_1$ and $h_2$ for the heights before and after the jump, respectively (the same for the velocities and densities). I'm left with the (bad) impression that using another notation may allow you to promote your approach but the approach is not so original at the end. Again, please see Gray et al. (J. Fluid Mechanics 2003), Hakonardottir and Hogg (Phys of Fluids 2005), Gauer and Johannesson (Handbook 2009, Chapter 11), Faug (Phys Rev E 2015) and Albaba et al. (Phys. Rev E 2018).

- page 2, lines 19-20: *"Because we predict the speed of the compaction front, and therefore the loading duration as a function of the incoming avalanche velocity, the method facilitates the use of dynamic magnification factors in structural analysis."*. There already exist relevant models, based on the correct equations (the shock-wave equations: see another comment below, on your eq. (4) which looks to be wrong by the way), to predict the speed of the traveling shock-wave, as proposed ealier for snow avalanches (see Chapter 11 of the EU Handbook (2009)) or studied in detail for granular flows impacting walls by Faug (Phys. Rev. E 2015) and Albaba et al. (Phys. Rev. E 2018), or also proposed for landslides interacting with dams (Iversion J Geophys Res 2016).

- Eq. (3): this equation needs more explanation. Should reads:

$$\frac{\mathrm{dK_\Phi}}{\mathrm{dt}} = \frac{\mathrm{d}}{\mathrm{dt}}(M_\Phi V_\Phi^2) = M_\Phi V_\Phi \dot{V}_\Phi + \frac{1}{2}\dot{M}_\Phi V_\Phi^2 \tag{1}$$

Could you explain why the first term is neglected? You are dealing with time-varying incoming flow conditions ($\dot{V}_\Phi$ should vanish under steady-state conditions only).

-page 4, lines 5-6: *"The difference between... is a measure of cohesion"*... What do you mean? Without any cohesion (in a dry granular flow) you also have a difference (=a jump).

- Eq. (4) is wrong: this equation proposed by the authors does violate the mass conservation across the shock-wave. Let us use some more standard notation in fluid dynamics: $\mathbf{U}$ is the speed of the traveling wave, and $f_1$ and $f_2$ for any variable before the shock and after the shock, respectively, and $[[f]] = f_2 - f_1$ the difference between the enclosed function $f$ on the forward and rearward sides of the singular shock surface. The correct equation in its depth-averaged form is:

$$[[\rho h(\mathbf{u} - \mathbf{U}) \cdot \mathbf{n}]] = 0 \tag{2}$$

This yields ($U < 0$, and $u_2 = 0$ in the dead zone against the wall):

$$U = -\frac{u_1}{X - 1}, \qquad (3)$$

if we note $X = \frac{\rho_2 h_2}{\rho_1 h_1}$. Your Eq. (4) gives $U = -\dot{S}_\Phi = -\frac{u_1}{X}$, which does not give the correct jump condition.

- Eq. (6): why this $1/2$ factor? This equation is false. Either it is a factor one and a difference in the brackets (sign $-$) or a factor $1/2$ and a sum (sign $+$).

- from now on, I'm a bit worried because Eq.(3), (4) and (5) are wrong or pulled out of a hat... Several equations in the rest of the paper should be corrupted by the mistakes made... and the plots (virtual cases shown and practical application) may be false too.

- (see previous comment) in particular, Eq. (10) clearly produces a drag coefficient which is wrong and does not satisfy the shock-wave conditions (see for instance Albaba et al. Phys. Rev E 2018).

- figure 2, caption: this is weird to state $h_2 = h1$ while you have a jump (you are talking about pile-up; see also a comment by Referee 1). Note that the density ratios used should merit some discussion. This is not an easy question in practice.

- figure 3: are you sure that you get $V_\Phi$ (the incoming flow velocity upstream of the obstacle) at the two outgoing sections downstream of the obstacle? You should write mass conservation before infering such a statement.

- page 8, line 9: why this (very personal) comment into brackets? Please remove it. I am sure there are scientists who are much more optimistic and will find a way of measuring this one day!

- page 8, lines 11-12: I agree that more data on snow avalanches and interaction with obstacles should be measured, about dead zone dynamics (compaction, length and shape evolution). BUT, please, refer/not ignore the information already available from the literature about granular flow/wall interaction: see for instance the recent paper by Albaba et al. (Phys. Rev. E 2018) where the shock-theory is compared to the dead zone dynamics measured in very informative numerical simulations on dry granular flows. In want of any precise measurements with snow, this information is rich.

- pages 13, lines 26-30, Eq. (17): I do not understand this statement at all. It should be removed. If I use your Eq. (16) and your Eq. (14) (recalling the latter is false) and neglect the "tractive" force term (as you do along your example), I exactly get that Eq. (17). Again, please check Albaba et al. (Phys Rev E 2018). The model proposed based on shock wave theory (mass and momentum conservation equations) gives also this form but the terms $C_D$ and $A$ are not constant and

(morever) coupled: such a model is correct and powerful.

- conclusions: if the authors are able to correct the flaws in their equations, they should explain why an energy approach is superior to already existing formula based on shock-wave theory relying on mass and momentum conservations.

**3  Recommendation**

For all the reasons given above, I conclude that the paper in its present form is far from being suitable for publication to NHESS. It seems to me that referees 1 and 2 both provided rather positive reports, however. Unlike both referees 1 and 2, I have no other choice but to provide a negative feedback on the current paper proposed by Perry Bartelt et al. That said, I would be happy to read in the future a revised paper if the authors can make an effort to :

- (i) correct the wrong equations,

- (ii) better explain their assumptions,

- and (iii) demonstrate that their energy approach is superior to existing methods (a cross-comparison would be needed and not only the plots from the energy approach by the authors) if points (i) and (ii) are carefully addressed, first.

With my best regards,
Thierry FAUG

---

## Author Comment (AC2) · 24 Oct 2018

Dear Thierry,

We simply object to you calling our mass balance "wrong" or "flawed". Furthermore, we have serious problems with your treatment of the snow pile-up process as a "shock" wave.

Mass conservation for a "compactible" material has many solutions. Consider a square of length l1 and height h1 and density rho1, A = (l1 h1). The first square represents the incoming avalanche mass (velocity u1) and the second and third squares represent the pile-up mass in front of the obstacle (velocity u2 = 0). For an "incompressible" material (rho1 constant) mass balance is simply given by the choice of h2 and l2. There are an infinite set of combinations of h2 and l2 which satisfy the problem.

[Figure]

For a compressible material there are even more possibilities, because we have introduced a third parameter, the "compacted" density rho2. Depending on this value, we still have an infinite number of possible solutions for l2 and h2 that satisfy mass conservation. The solution to the pile-up problem in front of the obstacle therefore requires a constitutive postulate stating how the mass piles-up. We can place more mass in the length l2 or more mass in the height h2. Be aware, however, that the choice is important, because it determines the braking distance (in the flow direction), and therefore the force on the obstacle.

The choice of the mass distribution (l2 and h2) and the choice of compacted density (rho2) (which, by the way we CAN and SHOULD measure in experiments or avalanche case studies) determines the "wave" speed. What is this "wave"? Here we make an analogy to vehicle traffic on a highway. For us it is simply the speed the interface is moving that exists between the moving avalanche mass (moving cars) and the piled-up snow mass (non-moving cars). It is moving away from the obstacle in the upstream direction. There might be a jump in height, or there might not be. It depends on how the avalanche snow compacts. (In vehicle traffic there is seldom a change in pile-up height, and each car loses all its kinetic energy in the non-moving zone. The pile-up interface exists without a jump in height.). We denote the speed of the interface Sdot. This speed (in the direction l2) depends on the constitutive assumption for the geometry of compaction. Clearly, in our model, avalanche mass that is missing on one side of the interface is gained on the other side (proof by simple geometry).

More importantly, an analogy to wave mechanics or shock waves is severely misplaced. For compaction (which is NOT the same as compression of an elastic body) the density of the incoming mass (volume*density) is NOT the same as the one of the stopped (smaller volume*higher density) mass. The mass is the same. In a shock wave, energy is transported, which in our model is not possible. The total kinetic energy of the stopped mass is dissipated during the reduction of the speed (over the braking distance d). S is not the position of a determined volume of mass. It is the location of the boundary, where we have the "compacted" density.

There is no way, we could agree with the wave concept, for energetic reasons. Snow is in our case simply an ideal inelastic material. If you have a travelling wave -- be it a shock wave or an "ordinary" wave -- you transport energy, which you certainly do NOT DO, when just stopping the material, "destroying" all its kinetic energy (transforming into heat/deformation). This is why the analogy to shock mechanics breaks down.
In a shock model, mass balance is formulated to accommodate the transfer of energy. In the pile-up process -- the avalanche impact problem – this is not necessary if you consider the stopping process to be completely inelastic.

The shock analogy therefore leads to results that are simply not in agreement with observations: we have observed avalanche pile-up zones with h1 = h2. We certainly do not want to exclude this case from the many possible set of pile-up configurations.

Finally we note the power of treating avalanche snow as an ideal inelastic material. The work theorem states that the change in kinetic energy $\Delta K$ is equal to the product of the force $F$ (on the wall *from pile-up*) and the braking distance $d$ of the avalanche mass stopped by the wall,

$$\Delta K = F\, d$$

Using the standard definition for impact pressure $p = {F}/{A}$

$$p = \frac{1}{2}\, C_d \rho\, U^2$$

(where $C_d$ = stress intensity factor; $\rho$ = density; $U$ = approach velocity, $A$ = cross-sectional area) we arrive at an interesting re-definition of the stress intensity factor:

$$C_d = \frac{V/_A}{d} = \frac{l}{d}$$

where $V$ is the volume stopped by the wall. With $l = V/_A$. the stress intensity factor (for pile-up) is simply defined as the length of incoming mass scaled to the braking distance. In our model, if the obstacle stops the mass immediately (d=0), then Cd is infinite (as well as the wave propagation speed), meaning there is an infinitely large force acting on the obstacle. This is the correct result, supported by experiments where large (but finite) Cd values have been measured.

The approach of invoking the work-energy theorem thus opens many doors to consider different impact situations, including the interpretation of experimental results. Frankly, it is impossible to interpret experimental measurements without knowing $l(t)$ or $d(t)$. We think that it is very important for practicing avalanche engineers to be presented with basic, compelling and consistent explanations.

---

## Referee Comment (RC4) · T. Faug (Referee) · 25 Oct 2018

Dear Perry and co-authors,

I (still) regret to say that your mass conservation given by Eq.(4) in your paper is not correct.

Let us use your notation. You must consider the relative velocities $V_r = V - \dot{S}_\Omega$ when writing mass continuity in a control volume moving at speed $\dot{S}_\Omega$ (your sketch drawn in figure 1):

$$\rho_\Phi h_\Phi V_{r,\Phi} = \rho_\Omega h_\Omega V_{r,\Omega}. \tag{1}$$

It follows:

$$\rho_\Phi h_\Phi (V_\Phi - \dot{S}_\Omega) = \rho_\Omega h_\Omega (V_\Omega - \dot{S}_\Omega). \tag{2}$$

Because the material between the wall and the discontinuity traveling at speed $\dot{S}_\Omega$ is at rest (in the domain $\Omega$), we have $V_\Omega = 0$. This then gives:

$$\rho_\Phi h_\Phi (V_\Phi - \dot{S}_\Omega) = -\rho_\Omega h_\Omega \dot{S}_\Omega, \tag{3}$$

and:

$$\rho_\Phi h_\Phi V_\Phi = -\rho_\Omega h_\Omega \dot{S}_\Omega + \rho_\Phi h_\Phi \dot{S}_\Omega \tag{4}$$

The equation above corresponds to the correct mass conservation, and yields:

$$\dot{S}_\Omega = \frac{-V_\Phi}{\frac{\rho_\Omega h_\Omega}{\rho_\Phi h_\Phi} - 1}. \tag{5}$$

(By the way, note that your velocities $\dot{S}_\Omega$ and $V_\Phi$, as defined in figure 1, are of opposite sign)

The calculations above (for $h_\Omega/h_\Phi > 1$) also work for the case $h_\Omega = h_\Phi$ if you would like to describe compaction against the wall without flow expansion upward ($h_\Omega = h_\Phi$ could be a reasonable assumption at the impact with the wall. However, after the initial impact of the snow avalanche with the wall this looks very unlikely to me).

Your Eq.(4) is wrong: it has to be corrected.

Regards,
Thierry FAUG

---

## Author Comment (AC3) · 26 Oct 2018

Dear Thierry,

**OUR MASS BALANCE is RIGHT!**

We apply the most elementary relation of hydraulics to describe the pile-up process, simply the flux of incoming mass into a bucket is equal to the speed the surface is rising in the bucket (see picture below). You will have us believe that the speed the level in the bucket is rising is faster because a "shockwave" propagates against the incoming flow. You argue that because the two are moving against each other, the incoming mass flux is larger and therefore the growth of the piled-up mass increases faster than we feed it!

[Figure]

What you forget is simple: in our case the "shockwave" **is the mass flux**. In the theory of shockwaves, the speed of the shock is due to the exchange of kinetic and potential energy (elasticity). It therefore has another source. When the mass balance is the source of the "shock", then the "shock" cannot modify the source. Your speed, $u_n$, must therefore be independent of the mass flux. When this is so, your mass balance model is correct (for example, in the case when. $u_n$ is equal to the speed of sound). In our case, we have no shock – we simply have mass piling-up in a "bucket" and consider the speed the surface is rising. This is again where the analogy with "shockwaves" breaks down completely.

Frankly, the rise of the surface level in the bucket is independent of whether we sit on the surface or consider it from the observer at rest (simple Galilean transformation). Sitting on the interface, I see the wall moving backwards, moving with the speed of Sdot (or $u_n$) and therefore it seems as the particles would flow with the speed Sdot through interface, such that for the outside observer the mass stops. In your model you only acquire a surplus of mass (incoming Sdot h1 rho1) and forget the same amount is going into the pile-up region with the speed Sdot. Mathematically, your model of mass balance is equivalent to filling the bucket with a hose where the hose is submerged in the

bucket (-u_n A1). You always subtract this term (u_n A1 rho1) or (u_n h1 rho1) from the correct mass balance but never add it back.  This produces pile-up speeds that are too high. This is clearly not the pile-up process as we observe it.  Furthermore, the question arises where is the interface?

To be honest, the analogy to shockwaves cannot be used **at all** to describe the pile-up process of an inelastic material, what avalanche snow is. We don't wish to be impolite, but we simply cannot understand why something so simple as filling a bucket with mass should be treated so complicated. Again, the application of the work energy theorem and simply hydraulic mass balances are sufficient to describe the external pile-up forces on structures.

---

## Short Comment (SC2) · 27 Oct 2018

Regarding the debate about the correct form of equation (4) of the presented paper by Bartelt et al., the question traces back to a common problem in the study of geophysical phenomena: The time rate of change of a domain containing a discontinuity surface. It can be expressed by the Reynolds transport theorem for the case when a volume is intersected by a moving discontinuity surface. The solution of the problem leads to set of jump conditions for the various balance equations as Thierry Faug indicates. The description of the problem can be found in textbooks about continuum mechanics and the willing reader is referred to those, e.g.:

[Figure]

Ichikawa, Y. & Selvadurai, A. P. S. Transport Phenomena in Porous Media, Springer Berlin Heidelberg, 2011 , 9-76, Chapter 2.

Hutter, K. & Jöhnk, K. Continuum Methods of Physical Modeling Springer, 2004, Chapter 3.

Greve, R. Kontinuumsmechanik Ein Grundkurs für Ingenieure und Physiker, Springer Berlin Heidelberg, 2003; Kapitel 2.2

Casey On the derivation of jump conditions in continuum mechanics International Journal of Structural Changes in Solids – Mechanics and Applications, 2011 , 3 , 61-84

or others.

Best Regards

Peter Gauer

---

## Author Comment (AC4) · 30 Oct 2018

The comment was uploaded in the form of a supplement: https://www.nat-hazards-earth-syst-sci-discuss.net/nhess-2018-154/nhess-2018-154-AC4-supplement.pdf

---

## Author Comment (AC5) · 30 Oct 2018

**Where is the mass that has velocity zero?**

Dear reviewers, editor and interested readers,

We thank all three reviewers for their positive and critical comments. The suggestions to improve the paper from positive reviewers 1 and 2 will be used to amend the text. The third reviewer, Thierry Faug, has stated bluntly that our mass balance equation is wrong, because it does not agree with the "shockwave" treatment of avalanche impact on a rigid structure. We are sincerely thankful for Thierry Faug's comments, because it forced us to review the "shockwave" approach. After our review of Thierry Faug's comments, we could identify the differences between our approaches and show why our method (application of the work-energy theorem) is correct, but also why we simply do not understand the "shockwave" treatment, since it has no physical relationship to the pile-up process.

The shockwave approach begins by considering the mass balance of an elastic wave travelling at the speed of u_n. Our approach is based on the idea of a kinematic wave[1], travelling with a speed equal to the pile-up rate of the avalanche snow in front of the wall. We denote the pile-up speed as Sdot. In our approach, there is no elasticity and therefore no energy exchange between the kinetic and potential (strain) energy driving the speed of the shockwave. We make no references to the speed of sound. Avalanche snow is considered to be an ideal, compactible, plastic material. The difference in this characterization between a pile-up and a "shockwave" leads to a fundamental change in the definition of the speed of the pile-up or "shock". For us, a "shock" simply cannot exist. It is a compaction of the mass in front of the wall.

Thierry Faug maintains that the correct balance is given by the formula

$$u\_1 \, (rho1 \, h1) = u\_n \, (rho2 \, h2 - rho1 \, h1)$$

which is the same as writing

$$u\_n = \frac{u\_1}{X - 1} \; with \; X = \frac{rho2 \, h2}{rho \, 1 \, h1}$$

Here, u_1 is the incoming avalanche speed, u_n is the "shock" wave speed propagating backwards, opposite to u_1. During the impact the snow mass compacts from the incoming density rho1 to the compacted density rho2. The incoming height of the avalanche is h1, the pile-up height is h2. Again, we emphasize, that this approach is correct for an elastic wave propagating with the speed u_n. However, it is incorrect for a kinematic or "pile-up" wave. We have no source for an elastic wave.

Consider the first time interval dt when an avalanche with height h1 hits the wall. Consider further that in this case the pile-up height doubles h2 = 2 h1, but there is no compaction, rho2 = rho1. This is shown in the figure 1 below. For this case, the "shock" wave model predicts that the out-going "wave" speed is equal to the incoming speed of the avalanche u_n = u_1. Thus, for the interval dt,
* * *
[1] The term "kinetic" wave comes from the theory of traffic jams, see "Kinetic Theory of Vehicular Traffic", by Prigogine and Herman, Elsevier Publishing, New York, 1971. Indeed, we treat snow pile-up as a "traffic jam".

the distance that the "shock" travels back (u_n dt) is equal to the length the incoming mass travels towards the wall (u_1 dt). Moreover, the avalanche hits the wall, and then is immediately deflected, traveling "on top" of the incoming avalanche. A shear plane must therefore develop in the pile-up zone and the "shock" front breaks down. In fact, there is no pile-up, only a deflection of mass. Moreover, the starting assumption of the shockwave model (a "shockwave" of height h2 propagates backward) is violated. It simply cannot happen with u_n = u_1. With our calculated speed Sdot, the pile-up wave extends over the entire height h2. The entire mass (u_1 h1 rho1) is stopped in front of the wall. We ask a simple question: where is the mass with zero velocity? Where is the pile-up? For us, it is in front of the wall.

[Figure]

**Figure 1: An avalanche with density rho1 and height h1 hits a rigid wall with the speed u_1. At impact, the mass does not compact rho2 = rho1, but grows to twice the height h2 = 2 h1. In the time interval dt, the volume of mass that moves towards the wall is u_1 dt and the mass that moves away from the wall u_n dt. This can only happen if a shear plane develops between the incoming and outgoing mass. Moreover, the "shock" front breaks down. There is no "pile-up". The avalanche hits the wall and is directed backwards. During a pile-up, the kinematic wave speed Sdot is maintained over the entire height h2.**

Another example serves to demonstrate why the shockwave approach is not suitable to model a pile-up and therefore avalanche impact pressures. Consider the case where the height of the incoming and piled-up mass are equal h2 = h1. The avalanche snow, however, compacts to twice the incoming density rho2 = 2 rho1. Again the speed of the "shock" (according to Thierry Faug) is given by u_n = u_1. This case is shown in Figure 2.

[Figure]

**Figure 2: : An avalanche with density rho1 and height h1 hits a rigid wall with the speed u_1. At impact, the mass compacts to rho2 = 2 rho1. The height of the pile-up does not change h2 = h1. In the time interval dt, the volume of mass that moves towards the wall is u_1 dt and the mass that moves away from the wall u_n dt. The shockwave model predicts u_n = u_1. Because the outgoing mass is compacted, this can only happen if an empty space develops between the compacted mass and the wall. Again, there is no "pile-up". The shockwave mass balance predicts that the empty space is filled with mass of density rho1. How?**

The results from the "shockwave" approach are simply bizarre. The avalanche mass hits the wall and propagates backwards with the density $rho2$ to the location $u\_1\ dt$ (the end location of the incoming mass). Moreover the speed of the incoming mass is equal to the "shock" speed $u\_n = u\_1$. However, the outgoing mass is compacted (the density behind the shock has increased). Therefore, the length of the compacted mass must be reduced because the height remains the same. Thus, it appears that an empty space develops between the outgoing mass and the wall. The empty space, however, is, according to the shockwave model, filled with the the mass, $u\_n\ rho1\ h1$. This is incredible: mass must travel through the outgoing wave (or perhaps it simply jumps over it) and piles-up with the density $rho1$ in front of the wall! This is simply not our picture (or any suitable mathematical characterization) of snow pile-up. For us, this completely erroneous result is a direct consequence of subtracting the amount $u\_n\ rho1\ h1$ from $u\_n\ rho2\ h2$ in the mass balance equation. This is the term that Thierry Faug wants us to be included in our mass balance. What is the physical reason for this inclusion? Simply because it agrees with a "shockwave" model? Again, we ask the question: where is the mass with zero velocity?

The same exercise can be performed over and over with different compaction ratios or outgoing/incoming height ratios $h2/h1$. The result is always the same. The shockwave model predicts incoming mass reaches the wall, where it is deflected backwards. This is not the pile-up process as we observe it, or how we model it. The predicted speeds of the pile-up front are too high. The shockwave model predicts that there is no non-moving mass of snow in front of the wall; the shockwave model does not allow a "pile-up" with density $rho2$ and height $h2$ (and speed zero, $u\_2 = 0$).

Based on these arguments, we have come to the conclusion that the "shockwave" analogy to mass pile-up in front of walls is wrong. Above all, we dislike the use of an "elastic" theory to explain the deformation of a plastic material and what we consider to be an entirely irreversible process.

The consequences of this disagreement are significant, for it prevents a better understanding of avalanche impact pressures. Our interest in the snow impact problem is motivated by improving the impact pressure calculations in the simulation model **RAMMS**, as well as providing rational explanations to practitioners, especially during user workshops. Our study of the existing literature, both experimental and theoretical, led us to seek better methods to calculate (and explain) impact pressures. From our reading, it appears to us that three areas of confusion (and therefore contention) now exist within the avalanche dynamics community:

1) Lack of a consistent theoretical model describing the impact process. For example, why should the method to calculate impact pressures change when the avalanche impacts a wall or thin pylon? The model should automatically take into account the geometry of the impacted object. Again our model is based on the application of the work-energy theorem to simplify the complex deformation mechanics of snow. It is a step in this direction because different structural geometries induce different pile-up processes and therefore braking distances. The shockwave model cannot resolve the problem of how snow deforms behind a specific geometry. Plastic, irreversible pile-up is central to understanding impact forces on BOTH wide and thin objects.

2) Lack of information on impact times. A terrible mistake that is now being propagated in the avalanche community is to equate the "external" avalanche forces to the "internal" structural forces (see several papers in the recent ISSW 2018 proceedings where measured external impacts forces are used to calculate internal bending stresses of the construction, e.g. VdlS mast). The pile-up force is an external force. The internal forces account for the inertial forces, which depend on the mass of and mass distribution within the impacted structure. Large, short duration external forces may have no effect on heavy (large mass) structures. Key in the future engineering analysis will be then to identify the duration of the pile-up loading, because this will determine how a particular structure reacts to an impact. The answer to the question, "where is the pile-up mass" is central. Because the external pressures are large does not necessarily mean the internal stresses are likewise large. The converse is also true: we can have external forces that excite resonance in the structure, increasing the internal stresses well above the static (i.e. external = internal) loads.

3) Interpretation of impact pressure measurements. To understand impact measurements it is essential to understand the pile-up geometry and duration time. The measured forces depend on the speed of the pile-up and therefore the geometry of the measurement device, which controls the stopping process. Each measurement device will have different pressure factors $Cd = ld/d$ (ld = effective length of incoming mass/d=braking distance). Frankly, the interpretation of measurements is lacking a theoretical model indicating what data and how must be collected in front of the structure (pile-up geometries, densities, etc). Because this additional information is missing, it is impossible to interpret pressure measurements and develop a consistent theory of avalanche impact. Empirical formulas could be replaced.

We conclude that the application of the work energy theorem, and the correct calculation of "shockwave" propagation speeds with the corresponding impact duration times, is a small step to understanding the snow avalanche impact problem. Where is the mass with zero velocity?

---

## Referee Comment (RC5) · Anonymous Referee #4 · 31 Oct 2018

General comments I appreciate to read this paper that search to explain why large impact occurs even at low avalanche velocities.

However, as yet proposed by Peter Gauer and Thierry Faug, the theory proposed is very similar to what can be found in the Chapter 11 of "The design of avalanche protection dams", without a citation and without an explanation of the originality of this new work. I think that at least the explanation given in the Bartelt's answer (26 September 2018) should be introduced in the paper.

I would have expected a validation with more data collected in the field, or at least less simplification (in many cases the hypothesis h $\Omega$=h $\varphi$ is done, a rectangular dead zone

is proposed. . .).

In addition I would have appreciated a more detail where new equations are introduced.

Specific comments:

-Page 2 line 27: "kinetic energy" : do you mean "flux of kinetic energy"?

-Page 2 line 27: The additional assumption that the width of the flow is larger than the obstacle width should be underlined.

-Page 2 line 30: why in Eq. (3) $dK\varphi/dt=0.5\cdot dM\varphi/dt\cdot V\varphi 2+ M\varphi\cdot v\varphi\cdot dv\varphi/dt$ the term $M\varphi\cdot v\varphi\cdot dv\varphi/dt$ is omitted?

- Page 2 Figure 1. For a more understandable figure the z-axis could be added (In this way it is more clear where x=0 is). In addition since Eq. (12) is based on the presence of a slope angle the x-axis could be inclined. Hence $\Delta d\Xi\rightarrow\Omega(t)$ could be represented in the figure.

-Page 4 line 4: what do you mean with "stationary"? I think that it is not in the sense of $dS\,\Omega\,/dt=0$.

-Page 4 line 5: "remains smaller": this is an assumption

-Page 4 line 12: it is not clear to me from where this equation come from, in particular the value 1/2. In addition $S\Omega$ is written without the dot (in discordance with Eq.(6)?

-Page 4 line 17: it is not clear to me from where this equation come from.

-Page 5 Figure 2: it is not clear to me in which way the Froude number (for me $Fr=u/(gh)0.5$ ) is related to the density.

- Page 5 line 11: Do you mean that the "friction component" is the total $p\Omega$ or only $\mu\Omega g$ z ?

-Page 6 Figure 3: the picture is not so clear. The x-axis drawn in vertical is difficult to interpreted: I suggest to turn the Side View picture.
- Page 6: Eq. 13 should be better explained

- Page 7 lines 25-26: "approximately twice" is in discordance with "between 3 and 5"

-Page 7 lines 27-32: what's the $\Omega$ value?

-Page 8 Figure 4, on the right: I don't understand why the value with $\Omega < \varphi$ is plotted (For instance $\varphi$=480 kg/m3 and $\varphi$=400 kg/m3 . In this way we would have a decompression!)

-Page 8 line 4: seen the Figure maybe it is better to use pT(t)«pΞ(t)

-Page 9 line 12: It is for this reasons that no data of $\Omega$ are available?

-Page 9 lines 13: why the name "RAMMS" is not explicited?

-Page 10 Figure 6 on the right: what do you mean with "Total pressure?" please explicit (for istance pΞ+ pΞ )

-Page 10 Figure 7: the shadow doesn't allow to well see the picture

-Page 11 line 8: How much is high the bridge? This information can help the reader to understand better the problem.

-Page 11 lines 13 -14: "Calculated. . . 50 kPa)" is a repetition of "excess of the standard pressure formula (50 kPa)" at line 12

-Page 12 line 10: why do you continue to underline the specific case of h $\Omega$=h $\varphi$?

- Page 13 line 4: it is not clear to me in which way the Froude number (for me Fr=u/(gh)0.5 ) is related to the density.

-Page 13 lines 8-10: it would be nice to see what happen at the density simulated by RAMMS, if a wall is inserted directly as a DTM modification. The model is able to describe the snow compaction or numerical instabilities occurs?

- Page 13 lines 25-26: Why do you say that it is not possible to experimentally measure

the pile-up in a test site? I think that at least characteristics as the final S$\Omega$, the final density $\Omega$ can be easily measured.

-Page 13 line 27: A(h$\Omega$,$\mu\Omega$) is not explained

- It is not clear to me why you talk about pile-up but you suppose h $\Omega$=h $\varphi$. In addition the results are compared with the standard equation p=v2 used in Switzerland where a hstau=v2/(2g$\lambda$) is considered too. If this to heights corresponds at two different processes it could be explained.

- The reference "The design of avalanche protection dams" should be cited

Typing errors:

- Pag.1 line 21: "process: When" -> "process: when"

- Pag.2 line 7: "importance: The" -> "importance: the"

- -Page 8 Figure 4 caption: "givne"-> "given"

- Page 7 line 30: "coeffieciens -> "coefficients"

- Page 8 line 14: "model: To" -> "model: to"

- Page 9 caption Figure 5: "V$\varphi$(0)=25 m/s" -> "V$\varphi$(0)=26 m/s"

- Page 11 line 7: "caclculated" -> "calculated"

---

## Referee Comment (RC6) · T. Faug (Referee) · 1 Nov 2018

Dear Editor, dear other referees, dear Perry and co-authors, dear interested readers,

As I'm still not convinced by the mass conservation (Eq. (4)) proposed by Perry Bartelt and co-workers, please allow me the following comments:

1) Mass conservation given by the jump condition is the correct way to write mass conservation when a flow impacts a rigid wall that spans the entire width of the incoming flow. As a result of the impact of the flow with the wall, a discontinuity in both velocity and density (if $h_\Omega = h_\Phi$), and in height (if $h_\Omega > h_\Phi$), forms and propagates at speed $\dot{S}_\Phi$ upstream of the wall. I maintain that the mass conservation across the MOVING discontinuity reads as follows (see details in item 2 below):

$$\rho_\Phi h_\Phi V_\Phi = -\rho_\Omega h_\Omega \dot{S}_\Phi + \rho_\Phi h_\Phi \dot{S}_\Phi. \tag{1}$$

(I'm using the notation of the authors.)

2) The sketch provided by Perry Bartelt and co-workers to interpret such a jump in velocity (from $u_1$ to $u_2 = 0$), density (from $\rho_1$ to $\rho_2 > \rho_1$) and height (from $h_1$ to $h_2$), as drawn in figure 1 –mid panel– of Perry Bartelt and co-workers' response published on 30 October 2018, is misplaced. The correct sketch rather corresponds to the right panel of figure 1 provided by Perry Bartelt and co-workers' in their response published on 30 October 2018, with the (more classical) following notation: $u_1 = V_\Phi$, $u_2 = V_\Omega = 0$, $h_1 = h_\Phi$, $h_2 = h_\Omega$, $\rho_1 = \rho_\Phi$, $\rho_2 = \rho_\Omega$, and $U = \dot{S}_\Phi$ ($= -u_n$ in my initial referee report on the paper).

To write mass conservation in such a situation, it is important to note that there exist discontinuities in variables (velocity, density, height) and the singular surface (where the discontinuities appear) is MOVING. This produces extra terms in the Reynolds transport theorem (see for instance the valuable references provided by Peter Gauer in his last comment). The Reynolds transport theorem reads as follows:

$$\oint_S \rho \, u \, \mathrm{d}S + \frac{\partial}{\partial t} \iint_V \rho \, V \mathrm{d}t = 0. \tag{2}$$

In the case when $h_2 = h_1$, the variation of mass in a control volume $V$ (surrounding the discontinuity) during $dt$ is:

$$\frac{\partial}{\partial t} \iint_V \rho \, V \mathrm{d}t = \rho_2 U - \rho_1 U, \tag{3}$$

(note that the terms above are $\rho_i h_1 U$ in the case $h_2 > h_1$.),

and the balance of mass fluxes across the control surface $S$ is:

$$\oint_S \rho \, u \, \mathrm{d}S = \rho_1 u_1 - \rho_2 u_2. \tag{4}$$

(Note that the terms above are $\rho_i h_i u_i$ in the case $h_2 > h_1$, and because $h_2 \neq h_1$ you have to add the extra mass flux term $-\rho_2(h_2 - h_1)U$ to the sum above.)

The Reynolds transport theorem then gives:

$$\rho_1 u_1 - \rho_2 u_2 = -\rho_2 U + \rho_1 U. \tag{5}$$

Using $u_2 = 0$ (mass stopped against the wall), it becomes:

$$\rho_1 u_1 = -\rho_2 U + \rho_1 U. \tag{6}$$

(for the case $h_2 > h_1$, we have: $\rho_1 h_1 u_1 = -\rho_2 h_2 U + \rho_1 h_1 U$. The term $\rho_2 h_1 U$ appears twice: in the variation of mass during $dt$ but also in the sum of mass fluxes. Thus, that term $\rho_2 h_1 U$ does not appear in the mass balance at the end. But the term $\rho_1 h_1 U$ is still here.)

With the notation of the authors, this reads:

$$\rho_\Phi V_\Phi = -\rho_\Omega \dot{S}_\Phi + \rho_\Phi \dot{S}_\Phi. \tag{7}$$

3) Perry Bartelt and co-authors are pointing out the analogy to kinematic waves formed during traffic jam. Please allow me to note that the Rankine-Hugoniot relation for mass continuity given by Eq. (5) is largely used in studies about traffic jam to extract the speed $U$ at which the wave propagates: $U = \frac{\Delta(\rho h)}{\Delta \rho}$. See for instance:

P. I. Richards. Shock Waves on the Highway. Operations research, 4(1): 42–51, 1956).

M. Lighthill and G. Whitham, On kinematic waves. II. A theory of traffic flow on long crowded roads. Proceedings of the Royal Society of London: Series A, 229 (1955), 317–345.

Regards,
Thierry FAUG

---

## Author Comment (AC6) · 20 Nov 2018

Many thanks for the comments. Very helpful. Based on your remarks we changed "the dot" notation. Also many figures have been updated. Please see the supplements containing both point by piont answers as well as the modified paper containg a comparison between "pile-up" and "shockwave" models.

Please also note the supplement to this comment:
https://www.nat-hazards-earth-syst-sci-discuss.net/nhess-2018-154/nhess-2018-154-AC6-supplement.pdf

---

## Author Comment (AC7) · 20 Nov 2018

Many thanks for your comments. We followed your suggestions and redid the figures. We added an entire section concerning the differences between the pile-up and shockwave approaches. Please see supplement.

Please also note the supplement to this comment: https://www.nat-hazards-earth-syst-sci-discuss.net/nhess-2018-154/nhess-2018-154-AC7-supplement.pdf

2018-154, 2018.

---

## Author Comment (AC8) · 20 Nov 2018

**Avalanche Impact Pressures on Structures with Upstream Pile- Up/Accumulation Zones of Compacted Snow.   Reply to Comments of the Four Reviewers**

**General:  We thank all four reviewers for the careful reading of the manuscript. The most serious criticism came from Reviewer 3 (Thierry Faug) who maintains (1) the manuscript does not adequately cite previous work on "pile-up" based on the analytical shock approach and (2) the presented approach, especially the statement of the mass balance at pile-up, does not agree with the existing publications.**

**To address these comments we have:**

(1) **added an entire section detailing the difference between the application of the work-energy theorem and the "shockwave" approach, see "Comparison of the work-energy approach to "shockwave" models".**
(2) **We have cited the existing literature.  We have added  two figures and a comparison table.**

**Based on our results, we show there a significant differences between the two approaches and why we think the "shockwave" approach has serious deficiencies. We come to the conclusion that the "shockwave" approach cannot model avalanche "pile-up", rather a "stream" of avalanche snow that hits the structure and is reflected. Subsequently, it is not suitable to the problem of pressures induced by slow moving avalanches. We explain our reasoning using numerical examples to demonstrate the difference in the two approaches.  We emphasize that this is an extremely significant result and will have many consequences for avalanche mitigation. The analogy to the propagation of an "elastic" wave to describe pile-up, a completely plastic process, does not satisfy our conditions for energy conservation.**

**Based on these results we have reformulated the conclusions with more emphasis on the model comparison.  The majority of the original conclusions hold, but we wanted to stress the inadequacies of existing experimental set-ups.**

**Other major revisions included**

(1) **Rephrasing the abstract and introduction**
(2) **Added a notation Table with a list of all variables.**
(3) **Removing the "dot" notation to describe changes. We use only the large "Δ", for example for the change in kinetic energy, or change in momentum.**
(4) **Redid all the Figures, according to suggestions of the reviewers.**

**We modified the examples, but would be happy to include more modifications if necessary, should the reviewers request more clarification.**

**Anonymous Referee #1**

The discussion paper titled "Avalanche Impact Pressures on Structures with Upstream Pile-Up/Accumulation Zones of Compacted Snow" by Bartelt et al. proposes a very interesting approach for dealing with the interaction between snow avalanche flow and isolated obstacles. The model considers that the deceleration of the snow mass turns into impact pressure against the obstacle. The paper clearly states its limitations: in particular, the considered geometry is simple, and an ideal rectangular dead zone is assumed (which is not real for narrow obstacles). In addition, the obstacle is supposed rigid, thus the dynamic effects are not considered. Nevertheless, the proposed mechanical model is worthy of attention, with particular reference to the engineering problems in mountain areas.

**Specific comments**

1. The authors state that cohesive flows with strong bonding between the snow clumps have the property $h_- \_ h$. In this case, a compaction of the impacting mass occurs, rather than a pile-up. A short comment is probably expected. → We write, "Cohesive flows with strong bonding between the snow clumps reduce the ability of the avalanche at impact to increase the pile-up height" Yes, it is interesting that "compaction" and "pile-up" can have different meanings. For us, stopping mass is "pile-up", even if it does not raise the height.

2. Referring to the analytical model, as well detailed in the discussion paper, the region $\_$ has length $V_{(t)}\_t$, while the resulting pile-up zone has length $\_S (t)\_t$. The authors indicate the braking distance as $d_\!$ (p.4 line 11). From the sketch in Figure 1, it results that the mean braking distance is the distance between the centers of mass of the compacting and the pile-up zones, i.e. $d_\!(t) = 12hV_{(t)}\_t \square \_S(t)\_ti$:

Why do the authors adopt a different symbol for the braking distance in Eqn. (5), i.e. $\_d_\!$? It is expected that $\_d_\!$ is the variation of the braking distance at different times, say t and t + $\_t$. In addition, the authors should also clarify what do they intend with $d\_\_\!(t)$. It is expected that this term is the time derivative of

the braking distance, i.e., $d\_\_\!(t) = \lim_{\_t!0}\_d_\!\_t = \lim_{\_t!0}d_\!(t + \_t) \square d_\!(t)\_t$:

Can the author better explain what do they intend with braking speed? Is it the ratio between the braking distance and $\_t$? Probably, it would be better to indicate the braking speed with a symbol without the dot. → Yes, we agree. We deleted the dot d notation entirely from the paper. It is not necessary. It simplifies the algebra (because we are always dividing by dt). When talking about the braking we should keep it simple and talk only about the braking distance.

3. Observing Figure 3, it seems that the shear traction force is directed against the snow avalanche flow, i.e., a negative pressure is acting on the obstacle. Have I well understood? →The traction applies a force on the pile-up zone. This force must be taken-up by the obstacle. That is, there is an equal and opposite reaction on the obstacle because of the traction.

4. Can the author include some references about the lateral requirements resistance of bridge guardrails? European norms (say EN 1317) relate to performance classes based on impact speed, angle and vehicle mass, rather than impact loadings. →We concentrated on the Swiss norms which are distributed by the ASTRA. We are simply not familiar with EU norms. We will make a literature search when the other issues are cleared up.

5. Limiting the attention to the failure of the guardrail, any impact pressure larger than the one that caused the observed damage would cause the same damage. However, the presence of further elements that were not destroyed by the avalanche can help in estimating an upper limit of the impact pressure. Have the authors found other elements that can help in estimating an upper limit of the impact pressure? →Unfortunately not. Or it would be very speculative. We took pictures of the damage and made a plasticity analysis, but did not want to introduce that into the paper. However, if the reviewer insists we can provide more details of the plasticity analysis.

**Minor observations**
• $\_S$ in Eqn. (5) → CHANGED
• The paper "Formation of levees and en-echelon shear planes during snow
avalanche run-out" by Bartelt et al. dates back to 2012, rather than 2017. → CHANGED

**Anonymous Referee #2**

The manuscript "Avalanche Impact Pressures on Structures with Upstream Pile-Up/Accumulation Zones of Compacted Snow" by Bartelt et al. discusses an approach to calculate/estimate pressures arising from avalanches hitting stationary obstacles. It presents an interesting mechanical model, the theoretical basis thereof, applied examples and states the limits and shortfalls. The setup of the paper is clear and follows a logical structure. The model seems to be ready to be included in dynamical avalanche models once the necessary basic changes (like variable densities) are implemented.

This presents a very promising and needed approach to discuss impact pressures on obstacles.

Main comments:
- The compaction density Rho_omega lacks the necessary discussion. Since it is one of the main driving factors for the results (e.g. fig 2 / 4) it needs a better justification. Rho_omega is currently not (yet) available from models, and reliable observations are not easily accessible. The other parameters of this approach can be handled by models or observations. So how do the authors suggest to handle this important (tuning) parameter? In the paper the model is sometimes tested with three (arbitrary, if plausible) densities, and sometimes set to a fixed value (e.g. label fig 9). Especially in the case for the Mittelbeda avalanche it is (seemingly) picked at random. This needs to be better justified with observations, or at least the reasoning for this specific value needs to be shown.
→ Yes, this is a problem. The only really good measurements of the compaction density is contained in the work of Thibert, which we now cite in the text. He measured a compaction density at Col de Lautaret of 540 kg/m3. Perhaps the paper should be seen as a plea for more measurements of compaction densities. We really don't have an answer.
- Figures 1 and 3 need to be improved. Figure 1 has separate compacting zone CZ and avalanche core AC in the upper panel. Then in the lower panel CZ and AC are the same, however the v is denoted with t + delta_t. I somehow expect there to be a CZ(t + delta_t) and the same for AC. Or if the authors try to show the "steady" state reached at end of compaction, remove the CZ in the lower panel and make it more clear in the label. The right panel of figure 3 contains basically the same (simplified) information as figure 1, adding only information about stress. Presenting the side view in the same manner as the top view leads to confusion. I suggest either to include the information about stress in fig 1 and reference it, or rotate the right panel of fig 3 by 90 deg to make the "side view" clearer.

→ We have completely redone the figures according to the specifications of the reviewer. Furthermore, we have improved our notation. We have removed references to "steady" since it is a dynamic process.

Comments:
- On p.4 / l. 4 it is stated "The pile-up height is generally...". How do the authors come to this conclusion? On p.2 /l. 25 part of this is presented as an assumption...
→ Yes, it is an assumption (a good one, we have never seen a pile-up height smaller than the flow). We state that it is one of the many assumptions.
- I suggest moving sentence p.5 / l. 9-10 to the beginning paragraph of section 2. This would be beneficial to the reader wondering about other influencing factors right from the start of the discussion.
→ With all the changes, perhaps it reads better now.
Regarding SC1 by Peter Gauer: I suggest including a short remark about the discussed work in the introduction.
→ See above, entire section is now introduced.
- For easier readability I suggest moving p. 8 / l. 6-12 to the beginning of the section. The information that both cases "are motivated by... " observations is an important one. - P 9. / l. 5: Remove "somewhat". Very unspecific: either it is unusual or it is not unusual.
→ Removed

- Out of interest: what causes the drop in velocity in fig. 9. a) at approx. 42 seconds?
→ In the simulations this is the end of the avalanche. In the RAMMS model, lower flow heights are associated with higher friction and therefore a decrease in velocity.. The last paragraph in the paper is extremely important to us – such questions cannot be answered without better observations and measurements.

Minor:
- Label fig. 4: givne -> given → CHANGED
- P 9. / l. 6: betweem -> between → CHANGED
- P 9. / l. 9: "at" missing between density and elevation → CHANGED
- P 9. / l. 11: possibilty -> possibility → CHANGED
All (obvious) typos are in section 4.3, seems to be avoidable by autocorrect.

Reviewer 3.

All the comments of reviewer 3 are addressed in the section "Comparison of the work-energy approach to "shockwave" models. We have come to conclusion that (1) the mass balance of the "shockwave" models and (2) the momentum balance of "shockwave" models do not model the "pile-up" process. We present a detailed comparison. We have serious doubts that the "shockwave" approach is correct.

**1 General comment**

The topic addressed by the authors is of utmost importance. The calculation of the impact force of avalanches on obstacles when a uid-to-solid transition oc- curs, thus forming stagnant (quasi-static) zone upstream of obstacles and traveling jumps, is a challenging question. Although a number of signi_cant advances were made in the recent years (most of them are available along my report below), there remains a lot to do because of the complicated physics which takes place during dense ow/obstacle interaction. The present paper proposes an approach based on a (simple) \energy approach".
I
read in detail the ideas developed by the authors. I must say that I have a number of major concerns about the theoretical part proposed by Perry Bartelt et al. The main reasons are (at least) the following:

The paper is firrst of all{ not free of misconceptions:
- _ the approach proposed starts with some equations that are pulled out of a hat (Eq. (3) for instance) or even wrong (Eq. (4)); this poses a serious problem because all the results presented in the rest of paper (virtual cases and practical case) depend on those confusing or awed equations stated at the beginning of the description of the model.
- _ I also found a couple of misleading statements (see the section speci_c comments below).

Moreover, I must say that the present study does ignore a number of important works done before on the topic. I thus fully agree with the short comment earlier proposed by Peter Gauer on this weak point of the paper.

Unlike referees 1 and 2, I cannot provide a positive feedback on the present study (see section Recommendation at the end of the present report).

I have taken time to write a rather detailed review in order to explain where the outcomes of the previous studies were relevant and could (not to say should) have been considered. I really invite the authors to read and consider the e_orts made in the recent years by other researchers on the topic. In particular, they should demonstrate that their energy approach (ONCE CORRECTED) is superior to previous approaches mostly based on momentum conservation equation. At this stage, I did not go through the details of the examples/applications (section 4) because some equations used to draw the different plots and presented at the beginning of the paper are answed.

**2 Specific comments**

- abstract: "Existing methods to calculate snow avalanche impact pressures on rigid obstacles are based on the assumption of no upslope pile-up of snow behind the structure at impact." This sentence shall help the author to promote their model but it must be removed because this is a wrong statement! There are methods|already published|that make e_ort to address carefully the problem and the authors should not ignore them: see the references cited along my review

comments below.

- page 1, lines 18-19: why using this term \shape coefficient" for $C_D$? In uid mechanics $C_D$ is the \drag coe_cient". Here, the EU handbook edited by Tomas Johannesson et al. (2009) would merit citation. In particular, table 12.1 (page 107) provides recommendations for the values of $C_D$.

- page 2, lines 3-5: it is well-established that when granular flows impact walls, the formation of dead zones upstream of the obstacle is a key process that we need to include into the impact force calculation in order to predict the correct impact forces. There are a number of published papers on the problem that would merit to be mentioned here: please see (Faug et al., Phys. Rev E 2009) for walls overtopped by a steady granular flows, see (Chanut et al., Phys. Rev. E 2010; Faug et al., Phys. Rev. E 2011) for walls overtopped by a transient granular flows, and more recently Albaba et al. (Phys. Rev. E 2018) for semi-infinite rigid walls (no overtopping). Moreover, it is also a well-established fact for snow avalanches/obstacle interaction problems that dead zones and shock-waves travelling upstream are important physical processes: see, and please cite, the EU Handbook 2009 (already mentioned above), and Faug et al. (Ann. Glaciol 2010).

Also some other papers on the topic by B. Sovilla and co-workers, as well as by Shiva Pudasaini and co-workers would merit much more attention.

page 2, line 3 again: note that \cohesive avalanches" is not a necessary condition. Dry (cohesionless) granular flows also produce dead zones at the impact with wide obstacles (see the literature mentioned above).
2 Figure 1: notation used here and all along the paper is weird... your drawing is nothing else than a traveling jump that is classically observed in water and granular ows (and snow avalanches) when those flows transition from a supercritical to a

subcritical flow regime (for instance when those flows impact a wall). The difference between water flows and granular (or snow) flows is the fact that the granular (and snow) jumps may be compressible and accompanied by a shock in density, in addition to both velocity and height discontinuities. In general, in fluid mechanics the notation used is $h_1$ and $h_2$ for the heights before and after the jump, respectively (the same for the velocities and densities). I'm left with the (bad) impression that using another notation may allow you to promote your approach but the approach is not so original at the end. Again, please see Gray et al. (J. Fluid Mechanics 2003), Hakonardottir and Hogg (Phys of Fluids 2005), Gauer and Johannesson (Handbook 2009, Chapter 11), Faug (Phys Rev E 2015) and Albaba et al. (Phys. Rev E 2018).

- page 2, lines 19-20: "Because we predict the speed of the compaction front, and therefore the loading duration as a function of the incoming avalanche velocity, the method facilitates the use of dynamic magnification factors in structural analysis.". There already exist relevant models, based on the correct equations (the shock-wave equations: see another comment below, on your eq. (4) which looks to be wrong by the way), to predict the speed of the traveling shock-wave, as proposed earlier for snow avalanches (see Chapter 11 of the EU Handbook (2009))
or studied in detail for granular flows impacting walls by Faug (Phys. Rev. E 2015) and Albaba et al. (Phys. Rev. E 2018), or also proposed for landslides interacting with dams (Iversion J Geophys Res 2016).
- Eq. (3): this equation needs more explanation. Should reads:
$dK\_dt = d dt (M\_V_2\_) = M\_V\_\_V\_ + 12\_M\_V_2\_ (1)$
Could you explain why the first term is neglected? You are dealing with time- varying incoming flow conditions ( $\_V\_$ should vanish under steady-state conditions only).

-page 4, lines 5-6: "The difference between... is a measure of cohesion\... What do you mean? Without any cohesion (in a dry granular flow) you also have a difference (=a jump).

- Eq. (4) is wrong: this equation proposed by the authors does violate the mass conservation across the shock-wave. Let us use some more standard notation in fluid dynamics: U is the speed of the traveling wave, and $f_1$ and $f_2$ for any variable before the shock and after the shock, respectively, and $[[f]] = f_2 \square f_1$ the difference between the enclosed function f on the forward and rearward sides of the singular shock surface. The correct equation in its depth-averaged form is: $[[\_h(u \square U) \_ n]] = 0$ (2)

This yields (U < 0, and $u_2 = 0$ in the dead zone against the wall):$U = \square u_1 X \square 1$; (3)if we note $X = \_2h_2\_1h_1$. Your Eq. (4) gives $U = \square \_S \_ = \square u_1 X$, which does not give the correct jump condition.

- Eq. (6): why this 1=2 factor? This equation is false. Either it is a factor one and a difference in the brackets (sign $\square$) or a factor 1=2 and a sum (sign +). - from now on, I'm a bit worried because Eq.(3), (4) and (5) are wrong or pulled out of a hat... Several equations in the rest of the paper should be corrupted by the mistakes made... and the plots (virtual cases shown and practical application) may be false too.

- (see previous comment) in particular, Eq. (10) clearly produces a drag coefficient which is wrong and does not satisfy the shock-wave conditions (see for instance

Albaba et al. Phys. Rev E 2018).

- Figure 2, caption: this is weird to state $h_2 = h1$ while you have a jump (you are talking about pile-up; see also a comment by Referee 1). Note that the density ratios used should merit some discussion. This is not an easy question in practice.

- _Figure 3: are you sure that you get V_ (the incoming ow velocity upstream of the obstacle) at the two outgoing sections downstream of the obstacle? You should write mass conservation before infering such a statement.

- page 8, line 9: why this (very personal) comment into brackets? Please remove it. I am sure there are scientists who are much more optimistic and will find a way of measuring this one day!

- page 8, lines 11-12: I agree that more data on snow avalanches and interaction with obstacles should be measured, about dead zone dynamics (compaction, length and shape evolution). BUT, please, refer/not ignore the information already available from the literature about granular ow/wall interaction: see for instance the recent paper by Albaba et al. (Phys. Rev. E 2018) where the shock-theory is compared to the dead zone dynamics measured in very informative numerical simulations on dry granular flows. In want of any precise measurements with snow, this information is rich.
- pages 13, lines 26-30, Eq. (17): I do not understand this statement at all. It should be removed. If I use your Eq. (16) and your Eq. (14) (recalling the latter is false) and neglect the "tractive" force term (as you do along your example), I exactly get that Eq. (17). Again, please check Albaba et al. (Phys Rev E 2018). The model proposed based on shock wave theory (mass and momentum conservation equations) gives also this form but the terms $C_D$ and A are not constant and 4 (moreover) coupled: such a model is correct and powerful.

- conclusions: if the authors are able to correct the answer in their equations, they should explain why an energy approach is superior to already existing formula based on shock-wave theory relying on mass and momentum conservations.

**3 Recommendation**
For all the reasons given above, I conclude that the paper in its present form is far from being suitable for publication to NHESS. It seems to me that referees 1 and 2 both provided rather positive reports, however. Unlike both referees 1 and 2, I have no other choice but to provide a negative feedback on the current paper proposed by Perry Bartelt et al. That said, I would be happy to read in the future a revised paper if the authors can make an e_ort to :_ (i) correct the wrong equations,_ (ii) better explain their assumptions,_ and (iii) demonstrate that their energy approach is superior to existing methods (a cross-comparison would be needed and not only the plots from the energy approach by the authors) if points (i) and (ii) are carefully addressed first.

**Anonymous Referee #4**

General comments I appreciate to read this paper that search to explain why large impact occurs even at low avalanche velocities.
However, as yet proposed by Peter Gauer and Thierry Faug, the theory proposed is very similar to what can be found in the Chapter 11 of "The design of avalanche protection dams", without a citation and without an explanation of the originality of this new

work. I think that at least the explanation given in the Bartelt's answer (26 September 2018) should be introduced in the paper.

→ We have now introduced an entire section that explains the difference between the "shockwave" and pile-up approaches. Please see introduction of the reply.

I would have expected a validation with more data collected in the field, or at least less simplification (in many cases the hypothesis h =h ' is done, a rectangular dead zone is proposed: : :). → In the conclusions, we now highlight the problem with the experimental measurements.

In addition I would have appreciated a more detail where new equations are introduced. Specific comments: → We have added more explanation to the equations. Including removing equations that are not necessary.

-Page 2 line 27: "kinetic energy" : do you mean "flux of kinetic energy"?

→ Yes,, we truly mean *the flux* of kinetic energy arriving at the obstacle. We now state it so.

-Page 2 line 27: The additional assumption that the width of the flow is larger than the obstacle width should be underlined.

→ Added the sentence: The width of the flow is assumed to be larger than the width of the obstacle.

-Page 2 line 30: why in Eq. (3) $dK'/dt=0.5\_dM'/dt\_V'2+ M'\_v'\_dv'/dt$ the term $M'\_v'\_dv'/dt$ is omitted?

→ Yes. There is no change of mass M' during the compaction process. Added the sentence: All the incoming mass is piled-up. For example, we consider no "splashing" or mass deflection at the obstacle.

- Page 2 Figure 1. For a more understandable figure the z-axis could be added (In this way it is more clear where x=0 is). In addition since Eq. (12) is based on the presence of a slope angle the x-axis could be inclined. Hence $\_d\_!(t)$ could be represented in the figure.

→ Added z-axis in the figures. We state that the braking distance is in the direction of flow which could be inclined. We do not consider the change in potential energy (for now) and say so.

-Page 4 line 4: what do you mean with "stationary"? I think that it is not in the sense of $dS\ /dt=0$. → We mean that it has no velocity. To avoid confusions we delete. We call it the dead zone which means that it has no velocity.

-Page 4 line 5: "remains smaller": this is an assumption. Yes, we say we have no overtopping of the wall. We changed the formulation: "Because we do not have overtopping, we assume it remains smaller than the height of the obstacle…"

-Page 4 line 12: it is not clear to me from where this equation come from, in particular the value 1/2. In addition S is written without the dot (in discordance with Eq.(6)? We deleted the braking distance rate equation. The ½ comes from the location of the center-of-mass (which is ½ the length of Vdt and Sdotdt). A dot over S was missing in Eq. 5.

-Page 4 line 17: it is not clear to me from where this equation come from. The rate of braking equation has been deleted. It is not necessary.

-Page 5 Figure 2: it is not clear to me in which way the Froude number (for me $Fr=u/(gh)0.5$ ) is related to the density. We assume that higher Froude numbers are associated with lower flow densities. Much of the existing literature presents results showing the pressure coefficient CD increasing with decreasing Froude number. We attempt to explain why this is so.

- Page 5 line 11: Do you mean that the "friction component" is the total p or only $\_g$ z ? We mean the total pOmega disappears. We write, "This friction component disappears on a flat slope gx = 0, pOmega = 0. In this case only the traction friction on the side of the pile-up zone is acting.

-Page 6 Figure 3: the picture is not so clear. The x-axis drawn in vertical is difficult to interpreted: I suggest to turn the Side View picture. →Yes, we can rotate the side view picture. We will rotate when the editor tells us to proceed.

- Page 6: Eq. 13 should be better explained. → We have added some lines of explanation for the explanation of Eq. 13.

Assuming we have some velocity-squared drag (parameter $\tau_T$), we can calculate the tractive force $F_T$ on one shear plane,
\begin{equation}
F_T = \tau_T \rho_{\Phi} V_{\Phi}^2(t) \left [ S_{\Omega} h_{\Omega} \right ].
\end{equation}
The total tractive stress on the obstacle arises from two shear planes, we find,
\begin{equation}
p_T(t) = 2 \tau_T \rho_{\Phi} \frac{S_{\Omega}(t)}{w_{\Upsilon}} V_{\Phi}^2(t).
\end{equation}

- Page 7 lines 25-26: "approximately twice" is in discordance with "between 3 and 5"
Yes, we deleted the line with "approximately twice. It is not necessary.
-Page 7 lines 27-32: what's the value?
We think the problem is with the word equivalent. The value of CD is between 3 and 5.
-Page 8 Figure 4, on the right: I don't understand why the value with <' is plotted
(For instance '=480 kg/m3 and '=400 kg/m3 . In this way we would have a decompression!)
Yes, this has to do with the fact that on the figure on the right we change the height of the pile-up from 2 m to 2.5 m, which leads to a "decompression". In this region the results are not very realistic.

-Page 8 line 4: seen the Figure maybe it is better to use pT(t)«p_(t)
-Page 9 line 12: It is for this reasons that no data of are available? In part. This is the typical situation. Before we can arrive to perform some forensic science, much of the evidence has been destroyed.
-Page 9 lines 13: why the name "RAMMS" is not explicited?
Because we want to demonstrate that the procedure is independent of the numerical model. The model must provide the velocity and flow height with an idea what the incoming flow density is. We can introduce it, but for us it is not necessary and distracting.
-Page 10 Figure 6 on the right: what do you mean with "Total pressure?" please explicit (for istance p_+ p_ )
-Page 10 Figure 7: the shadow doesn't allow to well see the picture -Page 11 line 8: How much is high the bridge? This information can help the reader to understand better the problem. → We made the drone flights as fast as we could after the event. We don't have better pictures. Sorry.
-Page 11 lines 13 -14: "Calculated: : : 50 kPa)" is a repetition of "excess of the standard pressure formula (50 kPa)" at line 12
Yes, we deleted second line.
-Page 12 line 10: why do you continue to underline the specific case of h =h '? Because this will always lead to the maximum pressures. Any increase in the pile-up height serves to reduce the pressure.
- Page 13 line 4: it is not clear to me in which way the Froude number (for me Fr=u/(gh)0.5 ) is related to the density. The higher the Froude number (velocity) the more disperse the flow and therefore the lower the flow density. This is a general relationship, that holds over a large density range.
-Page 13 lines 8-10: it would be nice to see what happen at the density simulated by RAMMS, if a wall is inserted directly as a DTM modification. The model is able to describe the snow compaction or numerical instabilities occurs? → RAMMS calculates the change in density (fluidization) due to surface friction. That is, where the frictional processes are located at the basal layer. It does not (yet) calculate the change in density due to a rigid impact with walls. This requires more work, especially describing the deformation field and the energy dissipation. We must remain with analytical solutions for now.
- Page 13 lines 25-26: Why do you say that it is not possible to experimentally measure the pile-up in a test site? I think that at least characteristics as the final S, the final density can be easily measured. → Yes, but we don't have this information for many of the existing

measurements. When we do, see for example the work of Thiebert, then we have wedge shaped pile-up distributions.

-Page 13 line 27: A(h,_) is not explained. It is an empirical fit parameter, that has been suggested by several authors.
- It is not clear to me why you talk about pile-up but you suppose h =h '. In addition the results are compared with the standard equation p=v2 used in Switzerland
where a hstau=v2/(2g_) is considered too. If this to heights corresponds at two different processes it could be explained. Any kind of increase in the pile-up height from the incoming mass will lower the pressures on the wall. We are always interested in the maximum pressures which occur for h_Phi = h_Omega. That is why we don't consider the hstau for now.
- The reference "The design of avalanche protection dams" should be cited → CITED many times

Typing errors:
- Pag.1 line 21: "process: When" -> "process: when" → CHANGED
- Pag.2 line 7: "importance: The" -> "importance: the" → CHANGED
- -Page 8 Figure 4 caption: "givne"-> "given" → CHANGED
- Page 7 line 30: "coeffieciens -> "coefficients" → CHANGED
- Page 8 line 14: "model: To" -> "model: to" →CHANGED
- Page 9 caption Figure 5: "V'(0)=25 m/s" -> "V'(0)=26 m/s" → CHANGED
- Page 11 line 7: "caclculated" -> "calculated" → CHANGED

**Avalanche Impact Pressures on Structures with Upstream Pile-Up/Accumulation Zones of Compacted Snow**

Perry BARTELT[1], Andrin CAVIEZEL[1], Sandro DEGONDA[2], and Othmar BUSER[1]

[1]WSL Institute for Snow and Avalanche Research SLF, Flüelastrasse 11, 7260 Davos Dorf, Switzerland
[2]ETH Institute for Construction, 8903 Hönggerberg, Zürich, Switzerland

*Correspondence to:* Perry Bartelt (bartelt@slf.ch)

**Abstract.** We apply the work-energy theorem to develop a method to predict avalanche impact pressures on rigid walls. The method treats snow at impact as an ideal plastic material and therefore accounts for the accumulation and pile-up of compacted snow in front of the wall. We show why the proposed method differs significantly from existing theories which treat the pile-up process using analogies to elastic "shockwave" propagation. We calculate under what conditions pile-up leads to large impact pressures at low avalanche approach velocities. The induced pressure depends on the incoming avalanche flow density relative to the ultimate compaction density because this determines the avalanche braking distance and therefore the flow deceleration in the upstream direction. The pile-up/accumulation process induces two additional pressures: (1) the static pressure of the pile-up zone and (2) the tractive stresses operating on the shear planes interfacing the accumulated and still moving avalanche snow. We demonstrate the use of the model on two theoretical examples and one real case study. Finally, we discuss the consequences of the application of the work-energy theorem for the interpretation of experimental measurements of avalanche impact.

**1 Introduction**

Recent works investigating avalanche-structure interaction have underscored the need to develop better methods to predict avalanche impact pressures (Ousset et al., 2015; De Biagi et al., 2015). There appears to be growing evidence that the long established engineering formula to calculate impact pressure $p(t)$ (time $t$, avalanche flow density $\rho_\Phi(t)$, avalanche velocity $V_\Phi(t)$ shape coefficient $C_D$),

$$p(t) = \frac{1}{2} C_D \rho_\Phi V_\Phi^2(t), \tag{1}$$

is only valid for certain avalanche flow regimes (Sovilla et al., 2008; Baroudi et al., 2011). The formula under predicts measured values, particularly for slower moving avalanches in plug-flow or "gravitational" regimes (Sovilla et al., 2016). In practice the under prediction is usually compensated by applying shape coefficients $C_D > 2$.

Equation Eq. 1 is based on two important physical assumptions. The first assumption is that no avalanche mass accumulates behind the structure during the impacting process: when a moving avalanche hits the structure, the smashed snow fragments are assumed to be immediately removed from the impacted surface and re-entrained back into the flow (Bozhinskiy and Losev, 1998). Moreover, avalanching snow is treated as a fluid in which the flux of incoming snow is in balance with the rate of mass

removal at impact. When this condition is satisfied, the application of Eq. 1 is acceptable, i.e. for dry, cohesionless avalanches consisting of disperse agglomerations of snow particles. The formula is correctly applied to model powder avalanche interaction with thin structures, such as trees (Feistl et al., 2014; Bartelt et al., 2018a). It is clearly not valid for slow, dense, cohesive avalanches impacting objects where the interaction causes the avalanche to stop or pile-up in front of the structure. That is, when avalanching snow exhibits some solid behaviour. Many avalanche defense structures – such as dams and other flow obstacles – are purposely designed to induce this process to stop dense flowing avalanches (Barbolini et al., 2009).

The second assumption is of equal importance: the impacted structure is assumed to be perfectly rigid. The structure dissipates none of the incoming flux of kinetic energy in structural deformation energy, but dissipates it entirely at impact in the snow avalanche. This assumption quite often leads to an overestimation of the internal stress state of the structure, especially when the duration of the loadings $p(t)$ is short. Far more serious is that the formula can lead to an underestimation of the structural deformations and therefore an under prediction of the true internal stress state of the structure when the time duration of $p(t)$ is near the resonance frequency of the structure (Thibert et al., 2008; Baroudi and Thibert, 2009). The application of Eq. 1 must therefore be combined with dynamic magnification factors to account for the impulsive response of the structure when assessing the possibility of structural failure (Clough and Penzien, 1975). Structural analysis from avalanche impact therefore requires methods that quantify the duration of the impact loading.

The purpose of this paper is to develop a mechanical model to calculate avalanche impact pressures for cases when snow accumulates and piles-up at impact, forming a region of compacted avalanche snow in front of the obstacle. Unlike existing "shockwave"-type methods that treat this problem (Barbolini et al., 2009; Albaba et al., 2018), we apply the work-energy theorem to determine the deceleration of the avalanche snow arising from plastic deformation and compaction. The model therefore accounts for the solid-like behaviour of avalanching snow (Eglit et al., 2007; Faug et al., 2010). We calculate the dynamic impact pressure as a function of the avalanche flow density $\rho_\Phi$ relative to the ultimate compacted solid density $\rho_\Omega$. Avalanche deceleration is calculated based on how kinetic energy is dissipated in the compaction zone. The results reveal why impact pressures in dense plug-flow regimes can be much higher than impact pressures in disperse flow regimes for equal approach velocity. Because we predict the speed of the compaction front, and therefore the loading duration as a function of the incoming avalanche velocity, the method facilitates the use of dynamic magnification factors in structural analysis. Perhaps more importantly, the method makes no analogy to elastic wave propagation and therefore predicts no reflection of mass at impact. This difference between the "shockwave" and work-energy approaches are highlighted in a separate section.

**2  Pile-up/Accumulation Impact Pressure**

We consider a dense avalanche $\Phi$ with velocity $V_\Phi$, height $h_\Phi$ and bulk density $\rho_\Phi$ impacting a rigid structure (Figure 1). The structure of width $w$ is positioned at the position $x$=0; the positive $x$-direction defining the upstream direction of the pile-up. For simplicity we assume the avalanche strikes the structure with a mean depth-averaged velocity and density; that is, both variables are constant over the flow height defined in the $z$-direction, but can vary in the streamwise direction and therefore time. We do not consider the impact of the powder dust cloud $\Pi$. For now we assume that the height of the structure $h$ is higher

[Figure]

**Figure 1.** Avalanche impacts a rigid wall. The avalanche consists of a flowing core $\Phi$ and powder cloud $\Pi$. The core is moving at the speed $V_\Phi$. The impact with the rigid wall creates a pile-up zone $\Omega$ (no velocity). In the compaction zone $\Xi$, incoming avalanche mass is decelerated from $V_\Phi$ to zero. The pile-up front represents the boundary between the moving and non-moving avalanche snow.

[Figure]

**Figure 2.** Mathematical model. Side view of avalanche impact with pile-up and accumulation. The upstream zone is divided into three regions: the dense flowing avalanche $\Phi$, the compacting region $\Xi$ and the pile-up or accumulation zone $\Omega$. The avalanche arrives at time $t$ travelling with the velocity $V_\Phi$, bulk density $\rho_\Phi$ and flow height $h_\Phi$. Within the time interval $\Delta t$ compaction zone $\Xi$ develops in front of the structure with length $V_\Phi \Delta t$. A pile-up zone $\Omega$ with length $S_\Omega$ develops. The pile-up zone is increasing at the speed $\dot{S}_\Omega$. The braking distance of the mass in $\Xi$ is $d_{\Xi \to \Omega} = \frac{1}{2}\left[V_\Phi \Delta t - \dot{S}_\Omega \Delta t\right]$.

than the flow $h_\Phi$ i.e. there is no overtopping of the structure. The width of the flow is assumed to be larger than the width of the obstacle. A list of the notation is provided in Table 1.

We describe the pile-up process by considering avalanche mass immediately before and after the pile-up (Figure 1). All the incoming mass is piled-up. For example, we consider no "splashing" or mass deflection at the obstacle. The avalanche is divided into "compacting" avalanche snow (region $\Xi$, density not yet $\rho_\Omega$, velocity not yet zero, time $t$) and "compacted" avalanche snow (region $\Omega$, density $\rho_\Omega$, no velocity, time $t + \Delta t$). Measurements of compacted snow density are rare. Thibert et al. (2008) measured a pile-up density $\rho_\Omega$ of 540 kg m$^{-3}$ in front of the instrumented pylon at the French Col du Lautaret test site.

The avalanche mass arriving at the obstacle $M_\Xi$ and stopping within the time interval $\Delta t$ is

$$M_\Xi = \rho_\Phi h_\Phi w \left[ V_\Phi \Delta t \right]. \tag{2}$$

The corresponding change of kinetic energy $\Delta K_\Xi$ of the avalanche is therefore

$$\Delta K_\Xi = \frac{1}{2} M_\Xi(t) V_\Phi^2 = \frac{1}{2} \rho_\Phi h_\Phi w \left[ V_\Phi^3 \Delta t \right]. \tag{3}$$

The length of the compacted, pile-up zone is denoted $S_\Omega$, the height $h_\Omega$. The pile-up height might be larger than or equal to the incoming avalanche height $h_\Omega \geq h_\Phi$. Because we do not have overtopping, we assume it remains smaller than the height of the obstacle $h_\Omega < h$. The difference between $h_\Phi$ and $h_\Omega$ is a measure of the cohesion. Cohesive flows with strong bonding between the snow clumps reduce the ability of the avalanche at impact to increase the pile-up height, $h_\Phi \approx h_\Omega$ (Bartelt et al., 2012, 2015). Because of the incoming avalanche, the length of the pile-up zone is growing at the rate $\dot{S}_\Omega$; it is given by conservation of mass,

$$\dot{S}_\Omega = \frac{\rho_\Phi(t) h_\Phi(t)}{\rho_\Omega h_\Omega} V_\Phi(t). \tag{4}$$

During the pile-up, the region $\Xi$ of length $V_\Phi(t) \Delta t$ in the $x$-direction compacts, increasing the length of the compaction zone $\Omega$, see Fig. 4. The difference in the locations of the center-of-mass of the compacting zone $\Xi$ and the piled-up mass $\Omega$ defines the braking distance $d_{\Xi \to \Omega}$ over which the incoming mass must stop,

$$d_{\Xi \to \Omega} = \frac{1}{2} \left[ V_\Phi \Delta t - \dot{S}_\Omega \Delta t \right]. \tag{5}$$

The mean force on the obstacle $\bar{F}_\Xi$ is found by equating the work-done by the braking and the change of kinetic avalanche energy in the compaction zone $\Delta K_\Xi$,

$$\bar{F}_\Xi d_{\Xi \to \Omega} = \left[ p_\Xi h_\Omega w \right] d_{\Xi \to \Omega} = \Delta K_\Xi. \tag{6}$$

The impact pressure $p_\Xi$ is found assuming the force is applied uniformly over the impact area $h_\Omega w$. Therefore,

$$p_\Xi = \frac{h_\Phi V_\Phi}{h_\Omega \left[ V_\Phi - \dot{S}_\Omega \right]} \rho_\Phi V_\Phi^2 \tag{7}$$

and with the subsitution of the equation for mass conservation

$$p_\Xi = \frac{h_\Phi}{h_\Omega} \left[ 1 - \frac{\rho_\Phi h_\Phi}{\rho_\Omega h_\Omega} \right]^{-1} \rho_\Phi V_\Phi^2. \tag{8}$$

From which it is possible to define an equivalent pressure factor $C_D$ for the pile-up/accumulation regime,

$$C_D = 2 \frac{h_\Phi}{h_\Omega} \left[ 1 - \frac{\rho_\Phi h_\Phi}{\rho_\Omega h_\Omega} \right]^{-1}. \tag{9}$$

5    Note that the dynamic pressure factor becomes infinite when $\rho_\Phi h_\Phi = \rho_\Omega h_\Omega$. These values of equivalent $C_D$ are in agreement with measured values for all $\rho_\Omega > \rho_\Phi$, see Fig. 3, and compare to Sovilla et al. (2008, 2016). This result suggests that impact pressures of slow moving avalanches can be large if the density of the incoming avalanche is near the compaction density. It is also possible to physically interpret the pressure factor $C_D$. Substitution of Eq. 1 into the work-energy theorem (Eq. 6) leads to

10    $$C_D = \frac{V_\Phi \Delta t}{d_{\Xi \to \Omega}}. \tag{10}$$

The pressure factor is therefore the length of the compaction zone $\Xi$ relative to the braking distance $d_{\Xi \to \Omega}$.

**3   Comparison of the work-energy approach to "shockwave" models**

Other models of avalanche pile-up have been advanced to determine the impact forces on walls. The most notable of these are the so-called "shockwave" models which are discussed in the avalanche mitigation handbook (see Barbolini et al. (2009),

15    chapter 11), or in recent papers (Faug, 2015; Albaba et al., 2018). These analytical approaches are derived from an analogy with the theory of elastic wave propagation; that is, the "pile-up" wave is considered as a "shockwave" that travels upstream when the avalanche impacts a structure. On one side of the "shockwave" mass is piled-up (velocity zero) while on the other side, incoming avalanche snow arrives (velocity = $V_\Phi$). The incoming avalanche snow impacts the piled-up, stationary mass which transfers the impact force to the rigid wall.

20    There are important differences between the two model approaches that deserve attention. In the pile-up model presented here, avalanche snow is considered a completely plastic material. There is no elastic deformation that is transferred over the flow discontinuity (the pile-up front). The incoming kinetic energy of the avalanche is consumed completely during the pile-up process.

Application of the work energy theorem provides the mean deceleration $a_{\Xi \to \Omega}$ of the incoming mass during the pile-up

25    process,

$$a_{\Xi \to \Omega} = \frac{1}{2} \frac{V_\Phi^2}{d_{\Xi \to \Omega}}. \tag{11}$$

Since we assume the deceleration $a_{\Xi \to \Omega}$ is constant over the time interval $\Delta t = t_1 - t_0$, the velocity of the incoming mass decreases linearly from $V_\Phi$ to zero over the braking distance $d_{\Xi \to \Omega}$. The braking distance determines the change in momentum

**Table 1.** Notation table. Dimension and definition. See Fig. 1.

| Symbol | Unit | Defintion |
|---|---|---|
| $\Phi$ | Subscript | Avalanche core |
| $\Pi$ | Subscript | Powder cloud |
| $\Lambda$ | Subscript | Air |
| $\Gamma$ | Subscript | Splashing, avalanche pre-front |
| $\Omega$ | Subscript | Pile-up zone |
| $\Xi$ | Subscript | Compacting zone |
| **Coordinate system, time, obstacle** | | |
| $x, y, z$ | m | Coordinate system, $x=0$ location of obstacle |
| $t, \Delta t$ | s | Time, time increment |
| $w$ | m | Width of obstacle |
| $h$ | m | Height of obstacle, $z$-direction |
| **Avalanche core $\Phi$** | | |
| $V_\Phi$ | m s$^{-1}$ | Flow velocity in the $x$-direction |
| | | (Positive towards the wall) |
| $\rho_\Phi$ | kg m$^{-3}$ | Bulk density of core $\Phi$ |
| $h_\Phi$ | m | Flow height of avalanche core $\Phi$ |
| **Compacting $\Xi$ and pile-up $\Omega$ zones** | | |
| $M_\Xi$ | kg m$^{-2}$ | Mass in the compacting zone $\Xi$ |
| $\Delta K_\Xi$ | J | Change of kinetic energy in the compacting zone |
| $\Delta P_\Xi$ | kg m s$^{-1}$ | Change of momentum in the compacting zone |
| $d_{\Xi \to \Omega}$ | m | Braking distance |
| $a_{\Xi \to \Omega}$ | m s$^{-2}$ | Deceleration of mass in the compacting zone |
| $S_\Omega$ | m | Length of pile-up zone in front of wall |
| $\dot{S}_\Omega$ | m s$^{-1}$ | Speed of pile-up front |
| | | (Positive away from wall) |
| $\rho_\Omega$ | kg m$^{-3}$ | Density of pile-up zone $\Omega$ |
| $h_\Omega$ | m | Pile-up height |
| $X$ | | Compaction factor $X = \rho_\Omega h_\Omega / \rho_\Phi h_\Phi$ |
| **Forces and pressures** | | |
| $G, g_x, g_z$ | m s$^{-2}$ | Gravity, gravitational components |
| $F_\Xi, \bar{F}_\Xi$ | N | Dynamic impact force and mean impact force from compaction and pile-up |
| $\bar{F}_T$ | N | Mean shear force on boundary |
| $p_\Xi$ | Pa | Dynamic impact pressure from pile-up |
| $p_\Omega$ | Pa | Static pile-up pressure |
| $p_T$ | Pa | Shear traction on pile-up boundary |
| $\mu_\Omega$ | | Coulomb friction coefficient in pile-up zone |
| $\tau_T$ | | Sliding friction coefficient, shear traction pile-up zone |

[Figure]

[Figure]

**Figure 3.** a) Effective $C_D$ coefficient (Eq. 9) for different incoming avalanche densities $\rho_\Phi$ and three compaction densities $\rho_\Omega$. The flow height and pile-up heights are equal $h_\Phi = h_\Omega$. Large effective $C_D$ coefficients result when $\rho_\Phi \approx \rho_\Omega$. In this case compacting (braking) distances are short and impact pressures are large. b) The calculated $\rho_\Phi C_D$ are in agreement with values derived from full scale measurements, e.g. (Sovilla et al., 2008). For the sake of comparison to measured values we plot the calculated $\rho_\Phi C_D$ values with decreasing density to mimic increasing Froude numbers (higher Froude numbers correspond to lower flow densities). This produces the effect that effective pressure factors $C_D$ are higher for lower flow velocities.

of the avalanche $\Delta P_\Xi$ and therefore the mean, time averaged force on the wall $\bar{F}_\Xi$,

$$\Delta P_\Xi = \int_{t_0}^{t_1} F_\Xi dt = \bar{F}_\Xi \Delta t = M_\Xi a_{\Xi \to \Omega} \Delta t. \tag{12}$$

The change in momentum is taken-up entirely by the obstacle. The idea of a braking distance is fundamental to the concept of pile-up. Note that a braking distance $d_{\Xi \to \Omega} = 0$ implies an infinite deceleration and therefore an infinite force. However, in

5 the pile-up model the energy is finite. It is not possible to transfer infinite energy to the structure. Within the framework of the pile-up model a braking distance $d_{\Xi \to \Omega} = 0$ indicates that all incoming avalanche mass is stopped instantaneously at the wall. There is no compaction. A negative braking distance $d_{\Xi \to \Omega} < 0$ is non-physical. It implies that the incoming avalanche snow has been somehow reflected backwards.

The concept of a braking distance over which a mass decelerates is not included in the "shockwave" model. The "shockwave"

10 model calculates the change in momentum to be, see (Barbolini et al., 2009; Albaba et al., 2018):

$$\Delta P_\Xi = M_\Xi V_\Phi + M_\Xi \dot{S}_\Omega = (\rho_\Phi h_\Phi V_\Phi \Delta t) V_\Phi + (\rho_\Phi h_\Phi V_\Phi \Delta t) \dot{S}_\Omega. \tag{13}$$

[Figure]

For $V_\Phi$= 10 m/s, $\dot{S}_\Omega$ = 10 m/s with $h_\Omega$= 2$h_\Phi$ and $\rho_\Phi$= $\rho_\Omega$    For $V_\Phi$= 10 m/s, $\dot{S}_\Omega$= 5 m/s

**Figure 4.** Difference between the "shockwave" and pile-up models for the case $X$= 2. In the "shockwave" approach the momentum balance is similar to a mass of $M_\Xi = \rho_\Phi h_\Phi V_\Phi \Delta t$ that impacts the wall and is reflected back such that $V_\Phi = \dot{S}_\Phi$ ($r$ = 1). The braking distance $d_{\Xi \to \Omega}$ = 0. Thus, in the "shockwave" model there is no stationary, piled-up mass in front of the wall.

From which the mean force on the wall $\bar{F}_\Xi$ is found to be,

$$\Delta P_\Xi = \bar{F}_\Xi \Delta t. \tag{14}$$

Although the end result agrees with the pile-up approach, it is highly problematic. The momentum balance of the shockwave model indicates that mass exists, travelling with the speed $\dot{S}_\Omega$, moving away from the wall. Note that because $V_\Phi$ and $\dot{S}_\Omega$ have

5   opposite directions, the change in momentum $\Delta P_\Xi$ is the sum of the momentum associated with each velocity. The equation for the change in momentum is therefore equivalent to a ball of mass $M_\Xi$ impacting the wall with the speed $V_\Phi$ that is reflected backwards with the speed $\dot{S}_\Omega$. The velocity $\dot{S}_\Omega$ is no longer the speed of a massless pile-up front, but it now represents the speed that mass is reflected backwards relative to the wall. In this sense, the "shockwave" model, does not model pile-up, rather a stream of incoming mass that is reflected backwards at the speed $\dot{S}_\Omega$ (see Figures 4 and 5). In the "shockwave" model the

10   ratio $r = \dot{S}_\Omega/V_\Phi$ can be considered a collisional restitution coefficient. In fact, when $r$ = 1, there is no energy loss.

In the "shockwave" model the speed of the pile-up wave is given from mass conservation by

$$\dot{S}_\Omega = \frac{V_\Phi}{\frac{\rho_\Omega h_\Omega}{\rho_\Phi h_\Phi} - 1} = \frac{V_\Phi}{X - 1} \tag{15}$$

where $X$ is the so-called compaction factor, see (Albaba et al., 2018). A numerical example shows that when $X = 2$, $\dot{S}_\Omega = V_\Phi$ and $r = 1$ and therefore we have no energy loss. This can been seen in Figure 4, where we have $\rho_\Phi = \rho_\Omega$ and $h_\Omega = 2\,h_\Phi$. In

15   this case the incoming mass cannot fill the pile-up volume (because the densities are equal) and appears to jump up to the free space (above $h_\Phi$) and is, at the same time reflected by the wall. Because the flow of incoming mass is continuous, it appears as if the stream of incoming mass is reflected backwards. The mass is reflected backwards on top of the incoming stream. In the pile up model, the incoming mass is stopped and distributed over the height $h_\Omega$ (compare in Fig. 4). We have, for the pile-up

model,

$$\dot{S}_\Omega = \frac{V_\Phi}{X}. \tag{16}$$

In the work-energy approach presented here, there is no reflection of mass. All incoming energy is dissipated. A slight deflection is possible to raise the height of the pile-up from $h_\Phi$ to $h_\Omega$.

5      Both the pile-up and "shockwave" models assume the density of the incoming mass ($\rho_\Phi$) is increased to the pile-up density ($\rho_\Omega$). However, in the "shockwave" model only a part of the mass is truly compacted. The other part of the incoming mass is used to "fill-in" the void space of its own volume to create the density $\rho_\Omega$. The "shockwave" mass balance therefore leads to a higher "shock" velocity than the pile-up speed. Mathematically,

$$\underbrace{\rho_\Phi h_\Phi V_\Phi}_{\text{Incoming mass}} = \underbrace{\rho_\Omega h_\Omega \dot{S}_\Omega}_{\text{Compacted mass}} - \underbrace{\rho_\Phi h_\Phi \dot{S}_\Omega}_{\text{Fill-in mass}}. \tag{17}$$

10    In order to "fill-in" its own volume ($\rho_\Phi\ h_\Phi\ \dot{S}_\Omega$), the mass in the volume must be stopped such that the flowing mass from behind can fill in the void space. This process ends when the density of the volume ($h_\Phi\ \dot{S}_\Omega$) reaches compaction density $\rho_\Omega$. In the pile-up model, the velocity gradient is concentrated at the moving front between the piled-up mass and the incoming mass (see Figs 4 and 5). There is no fill-in, only compaction of the incoming mass. All the incoming mass is moving with the speed of the avalanche and is completely compacted by changing the volume from $\rho_\Phi\ h_\Phi$ to $\rho_\Omega\ h_\Omega$. During this time, the incoming

15    mass experiences a constant, mean deceleration $a_{\Xi\to\Omega}$.

To understand the difference in the calculated pile-up speeds between the two models we must be aware that the definition of the braking distance $d_{\Xi\to\Omega}$ is independent of the model approach,

$$d_{\Xi\to\Omega} = \frac{1}{2}\left[V_\Phi - \dot{S}_\Omega\right]\Delta t. \tag{18}$$

It is the distance between the center-of-mass of the compaction volume ($1/2\ V_\Phi\Delta t$) and the location of the piled-up snow ($1/2$

20    $\dot{S}_\Phi\Delta t$). For the pile-up model $\dot{S}_\Omega = V_\Phi/X$ and for the shockwave model $\dot{S}_\Omega = V_\Phi/(X-1)$ (both from mass conservation). Therefore, the braking distance for the shockwave model is

$$d_{\Xi\to\Omega} = \frac{1}{2}\left[\frac{X-2}{X-1}\right]V_\Phi\Delta t, \tag{19}$$

while for the pile-up model, it is as before

$$d_{\Xi\to\Omega} = \frac{1}{2}\left[1 - \frac{1}{X}\right]V_\Phi\Delta t. \tag{20}$$

25    In principle, for a given pile-up speed, the braking distances should be the same for both models. For clarity we tabulate the different braking distances as a function of the magnification factors $X = 1$ to $5$ (Table 2). Cases $X < 1$ are non-physical (no compaction to a smaller density than $\rho_\Phi$) and cases $X > 5$ are non-realistic ($h_\Omega > 5\ h_\Phi$, $\rho_\Phi = \rho_\Omega$). We note that each model predicts the same impact force for each $X$, but the braking distances differ significantly. Surprisingly the "shockwave" model produces *negative* braking distances for $X < 2$. That is, the pile-up center of mass has now moved back, past the starting

**Table 2.** Calculated braking distance using "shockwave" and pile-up models. The calculated values for the pile-up model are all $d_{\Xi \to \Omega} \geq 0$. The mass balance of the shockwave model produces elastic reflections ($d_{\Xi \to \Omega} = 0$) for the case $X = 2$.

| $X$ | "Shockwave" $d_{\Xi \to \Omega}$ (x $V_\Phi \Delta t$) | Pile-up $d_{\Xi \to \Omega}$ (x $V_\Phi \Delta t$) | Comments |
|---|---|---|---|
| 1 | $-\infty$ | 0 | "Shockwave" model: infinite braking distance |
| | | | Pile-up model: infinite force |
| 1.5 | -1 | 1/6 | "Shockwave" model: negative braking distance |
| | | | Pile-up model: positive braking distance (see Figure 5) |
| 2 | 0 | 1/4 | "Shockwave" model: zero braking distance, finite force |
| | | | Pile-up model: positive braking distance, finite force (see Figure 4) |
| 3 | 1/4 | 1/3 | "Shockwave" model: not all energy dissipated |
| 4 | 1/3 | 3/8 | "Shockwave" model: not all energy dissipated |
| 5 | 3/8 | 2/5 | "Shockwave" model: not all energy dissipated |

location of the center-of-mass of the incoming volume. For the case $X = 2$, the entire incoming kinetic energy has been reflected backwards, $r = 1$, see Figure 4; for the $X = 1.5$ it appears that energy has been inputted into the pile-up to drive the reflected mass backwards at a speed twice the incoming speed, $r = 2$, see Figure 5. This simply cannot occur during a "pile-up" when all the mass in front of the wall is stationary. Again, it appears that in the "shockwave" model mass is not piled-up, but reflected backwards, sometimes even without the loss of energy.

In summary the application of the work energy theorem satisfies the balance of momentum as well as the conservation of energy. The mass in the pile-up zone has no velocity after the pile-up. The "shockwave" model suggests that there is some mass with kinetic energy (and therefore momentum) at the end of the pile-up process. Evidently, in the shockwave approach, there exist situations in which no energy is dissipated at all. These situations do not exist in the pile-up model, in which all kinetic energy is dissipated by irreversible compaction.

**4   Mass Accumulation Induces Tractive and Static Pressures on the Obstacle**

The total pressure acting on the structure consists of an additional two parts: (1) the static pressure $p_\Omega(t)$ and (2) tractive pressures that develop on the shear planes between the moving and piled-up snow $p_T(t)$, see Fig. 6 and (Faug et al., 2010)

$$p = p_\Xi + p_\Omega + p_T. \tag{21}$$

"Shockwave" mass balance $d_{\Xi \to \Omega} < 0$      "Pile - up" mass balance $d_{\Xi \to \Omega} > 0$

For $V_\Phi = 10$ m/s, $\dot{S}_\Omega = 20$ m/s with $h_\Omega = 1.5 h_\Phi$ and $\rho_\Phi = \rho_\Omega$      For $V_\Phi = 10$ m/s, $\dot{S}_\Omega = 6.66$ m/s

**Figure 5.** Difference between the "shockwave" and pile-up models for the case $X = 1.5$. The shockwave model predicts a negative and non-physical braking distance $\Delta d_{\Xi \to \Omega} = -1$. The incoming mass is reflected back such that $V_\Phi = 2\,\dot{S}_\Phi$ $(r > 1)$. Therefore, in the "shockwave" model there is no stationary, piled-up mass in front of the wall. This is not the case in the pile-up model.

All three pressures vary as a function of the accumulation zone $S_\Omega$ and the speed it is growing $\dot{S}_\Omega$. The static pressure of the pile-up zone $\Omega$ is given by

$$p_\Omega = \rho_\Omega S_\Omega \left[ g_x - \mu_\Omega g_z \right] \qquad \text{for} \qquad g_x > \mu_\Omega g_z \tag{22}$$

where $g_x$ and $g_z$ represent the gravitational accelerations in the $x$ and slope perpendicular directions $z$, respectively. The
5   Coulomb parameter $\mu_\Omega$ characterizes the basal friction upstream of the structure. The impact pressure in the pile-up/accumulation regime, unlike the dynamic pressure computed with the standard formula, will depend on the slope angle, as well as the terrain features surrounding the structure. This friction component disappears on a flat slope, $g_x = 0$, $p_\Omega = 0$.

On the boundary between the moving and non-moving snow tractive stresses develop. These can only be described by assuming some constitutive relationship between the moving planes, as well as some deformation geometry of the dead zone.
10  For the ideal case of a rectangular dead zone (constant width $w$), the shear tractions are perpendicular to the structure, requiring no rotation of the shear components into the coordinate system of the obstacle. The side area over which the tractive stress operates is $S_\Omega h_\Omega$. Assuming we have some velocity-squared drag (parameter $\tau_T$), we can calculate the mean tractive force $\bar{F}_T$ on one shear plane,

$$\bar{F}_T = \tau_T \rho_\Phi V_\Phi^2 \left[ S_\Omega h_\Omega \right] . \tag{23}$$

15  The total tractive stress on the obstacle arises from two shear planes, we find,

[revised manuscript text omitted]
 derivation does not rely on any analogy to the propagation of an elastic shockwave (Barbolini et al., 2009; Faug, 2015; Albaba et al., 2018). The pile-up model treats the accumulation of snow in front of the obstacle as a completely plastic process. The proposed method is based on the application of the work-energy theorem and is shown to conserve mass, momentum and

[Figure]

**Figure 12.** Mittelbeda avalanche, Davos. a) Calculated avalanche arrival velocity $V_\Phi(t)$ and density $\rho_\Phi(t)$ b) For a cohesive flow $h_\Phi = h_\Omega$, the calculated peak pressure is more than five times the value predicted by the standard formula (red line). Similar results to the standard formula are obtained when the pile-up height is $h_\Omega = 3.0$ m. Compaction density: $\rho_\Omega$=500 kg/m$^3$.

energy. A striking difference between the two model approaches is that the prediction of the pile-up speed which is associated with the deceleration of the compacting mass. We find non-physical braking distances for the "shockwave" model ($d_{\Xi \to \Omega} \le 0$ for $X \le 2$). Braking distances for the work-energy approach are all positive ($d_{\Xi \to \Omega} > 0$) for magnification factors greater than zero $X > 1$.

5    The condition to create large dynamic pressures for slow moving flows is therefore intimately related to the plastic deformation of avalanche snow and the formation of a pile-up/accumulation zone at the upstream face of the impacted obstacle. In this case the induced pressure $p_\Xi$ cannot be represented by the Froude number, rather the ratio of the density of the incoming snow $\rho_\Phi$ to the ultimate compacted snow density $\rho_\Omega$. We find for $h_\Phi = h_\Omega$,

$$p_\Xi(t) = \left[1 - \frac{\rho_\Phi}{\rho_\Omega}\right]^{-1} \rho_\Phi V_\Phi^2(t). \tag{25}$$

10   High density flows (that is typically slow moving flows) will exert large pressures when the snow cannot be compacted. In fact, in the theoretical case $\rho_\Phi = \rho_\Omega$, the impact pressure is infinite, because the braking distance reduces to zero. The braking distance, and therefore the magnitude of the force exerted on the obstacle, is directly related to the densification of the avalanching snow. Fortunately avalanche snow is a compactible material and the impact pressures are finite. Increasing the pile-up height $h_\Omega$ will reduce the applied pressure. Thus, cohesive flows which exhibit strong material bonding will exert higher
15   impact pressures on structures. The explanation why flows with low Froude numbers exhibit correspondingly higher pressures

is that these flows are simply denser, and their slow movement facilitates the formation of a pile-up zone. It is reassuring that the equivalent $C_D$ values we calculate

$$C_D = 2\frac{h_\Phi}{h_\Omega}\left[1 - \frac{\rho_\Phi h_\Phi(t)}{\rho_\Omega h_\Omega}\right]^{-1}. \tag{26}$$

are directly comparable to values derived from experimental observations (Sovilla et al., 2008, 2016). Avalanche dynamics
5   models will need to predict streamwise variations in avalanche flow density in order to calculate impact pressures for the pile-up/accumulation regime (Buser and Bartelt, 2015; Bartelt et al., 2016).

Another important conclusion we make from our analysis is that when avalanche mass accumulates behind structures the impact pressure $p(t)$ can be generally expressed as a sum of three components,

$$p(t) = p_\Omega(t) + p_T(t) + p_\Xi(t). \tag{27}$$

10   These components are all interrelated by the geometry of the dead zone which defines both the magnitude of the static pressure $p_\Omega(t)$ as well as the location of the shearing interfaces and therefore the reaction to the tractive pressures $p_T(t)$. In this paper we have considered only one possible geometry: a backfill zone of constant width $w$ equal to the width of the impacted structure. Our analysis therefore reveals that the total pressure $p(t)$ in the backfill regime is a complex function of many time-dependent parameters (e.g. incoming avalanche velocity and density) as well as time-independent material parameters
15   describing avalanche snow, specifically the compaction density $\rho_\Omega$, the friction in the pile-up zone $\mu_\Omega$ and the tractive friction on the shear planes $\tau_T$. We purposely limited the physical description of each pressure component ($p_\Omega(t), p_T(t), p_\Xi(t)$) to a *single* constitutive parameter for each process ($\mu_\Omega, \xi_T, \rho_\Omega$). Moreover, even the most simple pressure calculations will require engineers to assume some displacement configuration of the backfill process. This will be difficult, see (Faug et al., 2010) for the example of wedge shaped back-fill regions.

20   Our final conclusion underscores the limits of on-going field investigations. Our analysis reveals that to validate theories of avalanche impact pressure, field experiments must gather three different types of data using dissimilar measurement devices and techniques. Firstly, pressures sensors must be used to measure the *external forces* that are applied by the avalanche to the structure at impact. Secondly, the *internal stress state* must be ascertained to understand the dynamic response of the structure from the impulsive loading, e.g. (Thibert et al., 2008; Baroudi and Thibert, 2009). To determine the internal stresses within
25   the structure (i.e. failure), dynamic magnification factors must be found which depend on the stiffness and mass distribution of the impacted body. Thirdly, information concerning the *pile-up and stopping process* is needed. This includes the time of formation, density and detailed geometry of the pile-up zone. Perhaps in the near future it will be possible to measure the flow deceleration directly at impact. For example, by placing inertial sensors in the flow (Caviezel et al., 2018). Without this triptych of information it is impossible to link the measured external forces to a specific compaction/deformation mechanism. With the
30   exception of the work of Thibert et al. (2008), the authors do not know any data set that meets all three requirements.

*Acknowledgements.* This work was performed within the framework of the joint Austrian-Swiss project bDFA, a study of avalanche motion beyond the dense flow avalanche regime. We thank the Austrian Academy of Science (ÖAW) for their financial support as well as the Austrian

research partners (Austrain Research Centre for Forests, Torrent and Avalanche Control and the Universtiy of Innsbruck). Further support is provided by the on-going WSL research program Climate Change and Mass Movements (CCAMM).

[revised manuscript text omitted]